# Learning Local-Global Contextual Adaptation for Fully End-to-End Bottom-Up Human Pose Estimation

## Abstract

This paper presents a method of learning *LO*cal-*Gl*O*bal *C*ontextual *A*daptation for fully end-to-end and fast bottom-up human *P*ose estimation, dubbed as *LOGO-CAP*. It is built on the conceptually simple center-offset formulation that lacks inaccuracy for pose estimation. When revisiting the bottom-up human pose estimation with the thought of "thinking, fast and slow" by D. Kahneman, we introduce a "slow keypointer" to remedy the lack of sufficient accuracy of the "fast keypointer". In learning the "slow keypointer", the proposed LOGO-CAP lifts the initial "fast" keypoints by offset predictions to keypoint expansion maps (KEMs) to counter their uncertainty in two modules. Firstly, the local KEMs (e.g. $11 \times 11$) are extracted from a low-dimensional feature map. A proposed convolutional message passing module learns to "re-focus" the local KEMs to the keypoint attraction maps (KAMs) by accounting for the structured output prediction nature of human pose estimation, which is directly supervised by the object keypoint similarity (OKS) loss in training. Secondly, the global KEMs are extracted, with a sufficiently large region-of-interest (e.g., $97 \times 97$), from the keypoint heatmaps that are computed by a direct map-to-map regression. Then, a local-global contextual adaptation module is proposed to convolve the global KEMs using the learned KAMs as the kernels. This convolution can be understood as the learnable offsets guided deformable and dynamic convolution in a pose-sensitive way. The proposed method is end-to-end trainable with near real-time inference speed, obtaining state-of-the-art performance on the COCO keypoint benchmark for bottom-up human pose estimation. With the COCO trained model, our LOGO-CAP also outperforms prior arts by a large margin on the challenging OCHuman dataset.

## 1 Introduction

### 1.1 Motivation and Objective

Human pose is highly articulated with large structural and appearance variations. 2D human pose estimation in images is a classic structured output prediction problem, and remains a challenging one in computer vision and machine learning. Human pose estimation has numerous applications such as people-centered image understanding, autonomous driving and Augmented Reality (AR). With the recent resurgence of deep neural networks (DNNs), the performance of human pose estimation has witnessed remarkable improvement [12, 3, 15, 22, 11]. This paper focuses on the deep learning based problem formulation.

There are two deep learning based paradigms for human pose estimation in the literature. The top-down paradigm consists of human detection and single human pose estimation in each detected human bounding box [12]. The bottom-up paradigm also includes two components: human pose keypoint detection and keypoint grouping [3]. The top-down paradigm often obtains better accuracy performance, but suffers from its inferior efficiency since the computational cost of the single human

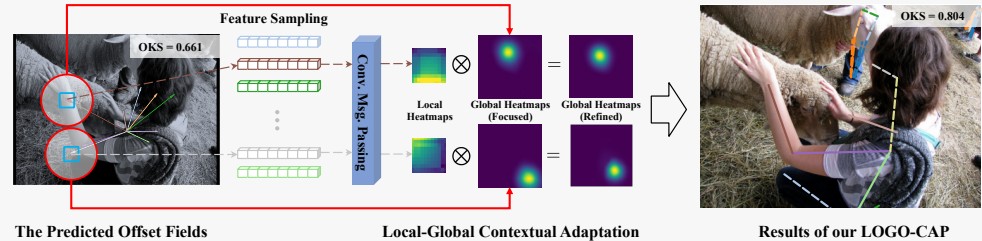

Figure 1: Illustration of the proposed LOGO-CAP for bottom-up human pose estimation. It is built on the center-offset representation. See text for detail.

pose estimation component is linearly scaled with respect to the number of detected human bounding boxes in an image. It is also largely affected by the performance of the human detection component (e.g., not handling occlusion very well). Thanks to its efficiency, especially in real-time applications, the bottom-up paradigm becomes more and more attractive. For both paradigms, state-of-the-art methods often are not fully end-to-end trained and utilize different post-hoc processing modules to improve the performance. This paper is interested in developing a fully end-to-end bottom-up paradigm and aims at bridging its performance gap with the top-down paradigm.

For the bottom-up paradigm, the recently proposed center-offset approach [6, 28, 26, 11] is a conceptually simple formulation (see the left of Fig. 1 for an illustrative example and Fig. 3 for the detailed workflow). It alleviates the need of sophisticated keypoint grouping. When introducing human keypoints centers (i.e., anchors) by treating objects as points [35], it encodes a human pose as a star structure using the offset vectors of human keypoints relative to the anchors. The main challenge of the center-offset regression paradigm lies in the difficulty of accurately learning offset vectors with large structural variations, especially the long-range ones, which also leads to inferior performance. This paper builds on the center-offset approach and addresses its drawback.

## 1.2 Method Overview

To address the drawback of the center-offset formulation, we build the intuitive idea of "**Keypointing, fast and slow**", by analogy to the modes of thought suggested by Daniel Kahneman in "*Thinking, fast and slow*" [14]: (i) *Fast Keypointer*: We treat the vanilla center-offset based estimation [35] as the *Fast Initializer* of pose estimation. (ii) *Slow Keypointer*: The lack of localization accuracy in the Fast Initializer entails a *Slow Solver* that learns to refine the "fast" keypoints. By slow, it is only relatively speaking. The Slow Keypointer is actually fast with near real-time speed.

To realize the Slow Keypointer, as illustrated in Fig. 1 and Fig. 3, this paper presents a method of learning **LO**cal-**GlO**bal **C**ontextual **A**daptation for fully end-to-end and fast bottom-up human **P**ose estimation, dubbed as **LOGO-CAP**. To quantitatively motivate the proposed method, we first present a surprisingly strong observation for a vanilla center-offset regression method (Table 1) in the fully-annotated subset of the COCO val-2017 dataset.Specifically, the vanilla regression method utilizes the HRNet-W32 [27] as the feature backbone to directly predict keypoints center heatmap and the offset vectors. This vanilla center-offset model obtains 60.1 average precision (AP), which is not great, but reasonably good. It

Table 1: The performance of a vanilla center-offset regression approach, its empirical upper bound, and the performance of our proposed LOGO-CAP using HRNet-W32 [27] as the feature backbone. See text for detail.

|  | Baseline | Emp. Bound | LOGO-CAP |
|---|---|---|---|
| AP | 60.1 | 88.9 | 70.0 |
| $AP^{50}$ | 85.2 | 93.1 | 88.2 |
| $AP^{75}$ | 66.7 | 90.6 | 76.4 |
| $AP^{M}$ | 53.7 | 87.7 | 64.4 |
| $AP^{L}$ | 71.5 | 90.2 | 78.4 |

clearly shows that the pose keypoints center and the offset vectors can be learned reasonably well. Instead of directly utilizing the learned offset vectors for human pose estimation, we treat them as human pose keypoint initialization and do a local window search to compute the empirical upper-bound of performance. More detailed, based on the predicted human poses, by introducing a local window (e.g., $11 \times 11$) centered at each detected key point and by computing the single keypoint similarity with the ground-truth keypoint, an empirical upper-bound of 88.9 AP is obtained, which is significantly higher than the state of the art and shows the potential of improving the vanilla center-offset regression paradigm.

Motivated by the above observation, a straightforward way is just to learn a local heatmap (e.g., $11 \times 11$) for each human pose keypoint based on the learned center and offset vectors, and then to

compute the refined keypoints by taking $\arg\max$ within the local heatmap. Although appealing, this does not work as observed during our development of the LOGO-CAP. The underlying reason is easy to understand: if this can work, the original offset vector regression should work at the first place since no additional information is introduced through learning the local heatmap. *We hypothesize* that on the one hand, on top of the local heatmap, the structural relationship between different keypoints of a human pose needs to be taken into account, and on the other hand, the intrinsic uncertainty of the local information in a local heatmap needs to be resolved. The former is the key challenge of structured output prediction problems. Many message passing algorithms have been developed in the literature. The latter can not be addressed by simply increasing the local window size. It entails learning stronger local-global information interaction and adaptation,.

Along with the two hypotheses, the proposed LOGO-CAP lifts the initial keypoints via the center-offset prediction to keypoint expansion maps (KEMs) to counter their lack of localization accuracy in two modules (Section 3.2). The KEMs extend the star-structured representation of the center-offset formulation to the pictorial structure representation [10, 8]. The first module computes local KEMs and learns to account for the structured output prediction nature of the human pose estimation problem, leading to the keypoint attraction maps (KAMs). The second computes global KEMs and learns to refine the global KEMs by leveraging the KAMs.

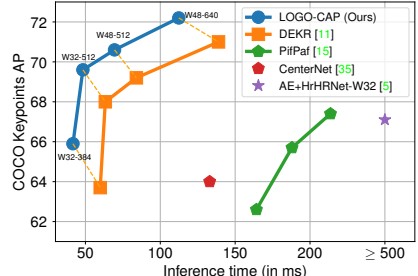

Figure 2: Speed-accuracy comparisons between our LOGO-CAP and prior arts on the COCO val-2017 dataset. W$x$-$Y$ (e.g. W32-384) means that a model uses the backbone HRNet-W$x$ (HRNet-W32) and is tested with the image resolution $Y$ in the short side.

Our LOGO-CAP is a fully end-to-end bottom-up human pose estimation method with near real-time inference speed. It obtains 70.0 AP in the fully-annotated subset of the COCO val-2017 dataset, which is an absolute increase of 9.9 AP compared to the vanilla center-offset method, making a significant step forward. Fig. 1 shows a pose estimation example. Fig. 2 shows the advantage of the proposed LOGO-CAP in terms of overall speed-accuracy comparisons between our LOGO-CAP and prior arts. Meanwhile, we should notice that there is also a significant gap compared to the empirical upper bound (Table 1), which encourages more work to be investigated.

## 2   Related Works and Our Contributions

There is a vast body of literature for human pose estimation. Many elegant representation schema have been developed for modeling articulated human pose in the traditional approaches such as the well-known pictorial structure model [10, 8] and its many variants [24, 1, 23, 33, 25]. Most of them focused on single person pose estimation. They perform inference over a combination of local observations on body parts (i.e., the data term) and the spatial dependencies between them (i.e., the spring or clique term). The spatial dependencies are captured either using directed and acyclic structures that facilitate the global optimization by dynamic programming [2, 9], or using structures with loop introduced (for high-order part relationship modeling) which resort to approximate inference by loopy belief propagation [19]. The bottleneck of the traditional methods lies in the data term which is often based on hand-crafted features. With the resurgence of DNNs and the end-to-end learning, the data term has been largely improved. We briefly review the recent deep learning based approaches for bottom-up human pose estimation.

**Limb-based Grouping Approaches** have been extensively developed due to the naturalness of modeling limbs based on keypoints. Given a predefined limb configuration (e.g., the COCO person skeleton template consisting of 19 limbs based on 17 keypoints), the grouping can be addressed by Part affinity field (PAF) [4, 3], Associative Embedding (AE) [20], mid-range offset fields in Person-Lab [22] and the fields of Part Intensity and Association [15]. Typically, sophisticated designs are entailed to achieve good performance. For example, a bipartite graph matching is used in Open-Pose [3]. In addition to be computationally expensive, another drawback of these methods is not fully end-to-end trainable. More recently, the differentiability issue was studied by the Hierarchical Graph Clustering (HGG) method [13], which utilizes graph convolution networks to repeatedly delineate pose parameters of multiple persons from a keypoint graph. HGG improves the performance compared to its baseline, the Associative Embedding method [20] at the expense of significantly

increased computational cost. In contrast to thoses approaches, our proposed LOGO-CAP is fully end-to-end trainable and achieves near real-time inference speed.

**Direct Regression based Approaches** have attracted much attention due to their conceptually simple formulation [6, 28, 26, 11, 30]. These center-offset based formulation are inspired by the recent remarkable success of direct bounding box regression in object detection such as the FCOS method [29] and CenterNets [35, 6]. As aforementioned, one main challenge is the difficulty of accurately regress the offset vectors, especially for the long-range keypoints with respect to the center. Sophisticated post-processing schema are often entailed to improve the performance. For example, a method of matching the directly regressed poses to the nearest keypoints that are extracted from the global keypiont heatmaps is used in [35]. Although being simple, the performance of this line of work is usually inferior to the limb-based approaches. The mixture regression network [30] alleviated the issue of regression quality to some extent, but still remained an indispensable performance gap comparing with the grouping-based approaches. Most recently, Geng *et al.* presented the first competitive direct method, DEKR [11] with a novel pose-specific neural architecture for disentangled keypoint regression. To improve the performance, the DEKR method utilizes a lightweight rescoring network to recalibrate the pose scores that are computed based on the keypoint heatmaps. Despite good performance, the DEKR method entails the additional rescoring stage in both training and testing, and thus is not fully end-to-end. The proposed LOGO-CAP retains the simplicity of the vanilla center-offset formulation and enjoys fully end-to-end training and fast inference speed.

**Our Contributions.** The proposed LOGO-CAP makes three main contributions to the field of bottom-up human pose estimation: (i) It addresses the drawback of the vanilla center-offset formulation while retaining its efficiency. It proposes the key idea of lifting a keypoint to a keypoint expansion map to counter the lack of localization accuracy. To our knowledge, it is the first fully end-to-end trainable method that achieves state-of-the-art performance. (ii) It presents a novel local-global contextual adaptation formulation that accounts for the nature of structured output prediction in human pose estimation and harnesses local-global structural information integration. (iii) It obtains state-of-the-art performance in the COCO val-2017 and test-2017 datasets. It also shows state-of-the-art transferability performance in the OCHuman dataset.

# 3 Approach

## 3.1 Problem Formulation

We follow the COCO protocol of defining the human pose. It consists of 17 human pose keypoints: 8 pairs of symmetric keypoints (hips, ankles, knees, shoulders, elbows, wrists, ears and eyes) and the nose keypoint. Let $P = \{1, \cdots 17\}$ be the set of keypoint indexes using a predefined order. Let $\Lambda$ be an image lattice of the spatial size $H \times W$ (e.g., $512 \times 512$), and $I$ be an image defined on $\Lambda$. Let $P_I^n$ be the set of keypoint indexes for a human pose instance $n$ in an image $I$ and we have $P_I^n \subseteq P$. For example, in COCO, we typically have $1 \leq n \leq 30$, and different human pose instances have different number of visible keypoints due to occlusion and/or truncation. Denote by $L_I^n = \{(x_i, y_i); i \in P_I^n\}$ the keypoint locations of a human pose instance $n$ in an image $I$, where $(x_i, y_i) \in \Lambda$. In the center-offset formulation, we introduce the keypoints center (i.e., the anchor), $(x_c, y_c)$ based on a given $L_I^n$ and we have,

$$x_c = 1/|L_I^n| \cdot \sum_{i \in P_I^n} x_i, \quad y_c = 1/|L_I^n| \cdot \sum_{i \in P_I^n} y_i. \tag{1}$$

With the anchor, a keypoint $(x_i, y_i)$ is equivalently defined by its offset/displacement, denoted by $(\Delta x_i, \Delta y_i)$ with $\Delta x_i = x_i - x_c$ and $\Delta y_i = y_i - y_c$. So, $L_I^n$ can also be equivalently expressed as $L_I^n = \{(x_c, y_c), (\Delta x_i, \Delta y_i); i \in P_I^n\}$.

The objective of human pose estimation is to recover $L_I^n = \{(x_i, y_i); i \in P_I^n\}$ for all human pose instances in an image. Denote by $\hat{L}_I^n = \{(\hat{x}_i, \hat{y}_i); i \in P_I^n\}$ the estimated human pose. Following the COCO protocol, the object keypoint similarity (OKS) is used to evaluate the accuracy,

$$\ell_{OKS}(\hat{L}_I^n, L_I^n) = 1/|P_I^n| \cdot \sum_{i \in P_I^n} \exp\left(-d_i^2/2s^2\kappa_i^2\right), \tag{2}$$

where $d_i$ is the Euclidian distance between the ground-truth keypoint $(x_i, y_i)$ and the predicted one $(\hat{x}_i, \hat{y}_i)$. $s$ is the square root of the human segment area, and $\kappa$ per-keypoint constant that controls fall-off in evaluation. We have $\ell_{OKS}(\hat{L}_I^n, L_I^n) \in [0, 1]$. The OKS metric is to evaluate the distance between predicted keypoints and ground-truth keypoints normalized by the scale of the person with the importance of keypoints equalized. In benchmarking different methods, the average precision

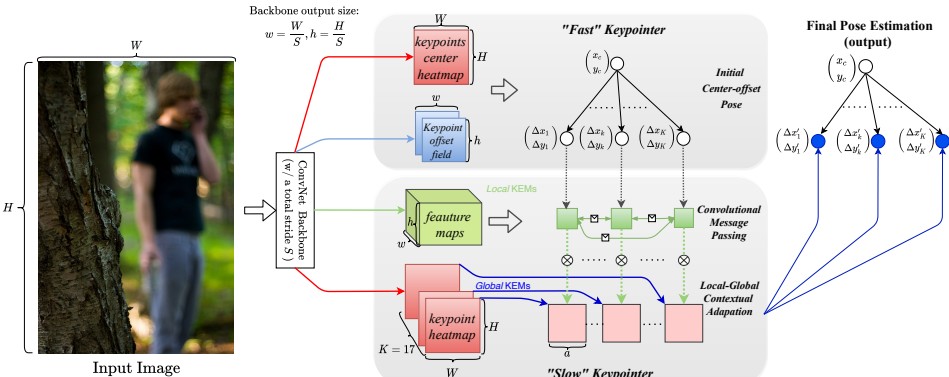

Figure 3: Illustration of the network and algorithmic flow of the proposed LOGO-CAP for bottom-up human pose estimation. See text for detail.

(AP) at OKS= $0.50 : 0.05 : 0.95$ is used as the primary metric, together with $AP^{50}$ at OKS= $0.50$, $AP^{75}$ at OKS= $0.75$, and AP across medium and large scales, $AP^M$ and $AP^L$ respectively.

## 3.2 The Proposed LOGO-CAP

We first present the network and the inference of LOGO-CAP, and then give details of the training. We keep different modules of the proposed LOGO-CAP simple, which in turn highlights the effectiveness of the proposed representation and algorithmic flow.

### 3.2.1 The Network and the Inference

As illustrated in Fig. 3, the proposed LOGO-CAP consists of four components as follows.

**i) A convolution neural network feature backbone.** Given an input image $I$, the output of the feature backbone is a $C$-dim feature map, denoted by $F \in R^{C \times h \times w}$, where $C$ is the feature dimension of the last convolutional layer in the feature backbone, and the spatial size $h \times w$ depends on the total stride in the feature backbone. We use off-the-shelf HRNets [27] in our experiments.

**ii) A parallel keypoint-offset regression module.** Given the feature map $F$, the output of keypoint regression is an 18-dim feature map (i.e., heatmaps) for the 17 keypoints and the keypoints center respectively. Denote by $\mathcal{H} \in R^{18 \times h \times w}$ the heatmaps, and by $\mathcal{H}^\uparrow \in R^{18 \times H \times W}$ the up-sampled heatmaps (using bi-linear interpolation in our experiments). The output of offset regression is a 34-dim feature map (i.e., the offset vector fields) for the 17 keypoints. Denote by $\mathcal{O} \in R^{34 \times h \times w}$ the offset fields. We adopt a minimally-simple design in realizing the regression modules using a channel-wise multi-layer perceptron (MLP). In implementation, we first apply dimension reduction to the feature map $F$ using a $1 \times 1$ convolution followed by a Batch Normalization (BN) and a Rectified Linear Unit (ReLU). Then, the output is computed by a $1 \times 1$ convolution. More specifically, we have the two parallel branches as follows,

$$F_{C \times h \times w} \xrightarrow[C \times 1 \times 1 \times C_1]{Conv+BN+ReLU} F^{\mathcal{H}}_{C_1 \times h \times w} \xrightarrow[C_1 \times 1 \times 1 \times 18]{Conv} \mathcal{H}_{18 \times h \times w} \xrightarrow[\text{bi-linear}]{UpSampling} \mathcal{H}^\uparrow_{18 \times H \times W}, \quad (3)$$

$$F_{C \times h \times w} \xrightarrow[C \times 1 \times 1 \times C_2]{Conv+BN+ReLU} F^{\mathcal{O}}_{C_2 \times h \times w} \xrightarrow[C_2 \times 1 \times 1 \times 34]{Conv} \mathcal{O}_{34 \times h \times w}, \quad (4)$$

where $C_1$ and $C_2$ are predefined (e.g., $C_1 = 32$ and $C_2 = 256$ are typically used).

*Initial pose estimation via the center-offset approach.* Based on the computed keypoints center heatmap $\mathcal{H}^\uparrow_{(18)}$ and offset fields $\mathcal{O}$, a predefined maximum number of pose candidates is computed as done in the vanilla center-offset approach. A non-maximum suppression (NMS) with a $3 \times 3$ window is applied in $\mathcal{H}^\uparrow_{(18)}$ and then the top-$N$ keypoints centers are selected (e.g., $N = 30$ in our experiments). The $N$ pose instances are computed by retrieving their offset vectors in $\mathcal{O}$ based on the selected $N$ keypoints centers. The $N$ pose instances are further pruned by thresholding their confidence scores in $\mathcal{H}^\uparrow_{(18)}$ with a predefined threshold (e.g., $0.01$ used in our experiments). Without confusion in the context, we still use $N$ to denote the number of poses instances by this initial pose estimation step. We obtain the set of estimated keypoints centers, denoted by $\mathcal{C}_{N \times 3}$ each row of which represents the position coordinates and the confidence score.

**Lifting a keypoint to a keypoint expansion map (KEM) by imposing a mesh.** For each of the $N$ pose instances, each of the 17 keypoints are placed in a local geometric mesh (e.g., $11 \times 11$) with the estimated location as the mesh center, capturing the uncertainty of the center-offset pose estimation as aforementioned in the introduction. This mesh can thus be interpreted as keypoint expansion map (KEM), accounting for competency-aware representations. The entire mesh is denoted by $\mathcal{M}_{N \times 17 \times 11 \times 11 \times 2}$, which is used in computing the empirical upper bound in Table 1. We have,

$$\{\mathcal{H}^{\uparrow}_{(18)}, \mathcal{O}_{34 \times h \times w}\} \xrightarrow[\text{center-offset}]{\text{initial pose estimation}} \{\mathcal{C}_{N \times 3}, \mathcal{M}_{N \times 17 \times 11 \times 11 \times 2}\} \tag{5}$$

**iii) A convolution message passing module.** We first encode the geometric mesh $\mathcal{M}_{N \times 17 \times 11 \times 11 \times 2}$ in a latent space with the dimensionality $C_3$ (e.g., $64$ in our experiments), computed based on the feature backbone output. Then, a keypoint is represented by a $C_3 \times 11 \times 11$ local feature map. A pose instance is represented by concatenating all the 17 keypoints. We have,

$$F_{C \times h \times w} \xrightarrow[C \times 1 \times 1 \times C_3]{Conv+BN+ReLU} F^{\mathcal{M}}_{C_3 \times h \times w} \xrightarrow[\text{bi-linear}]{\mathcal{M}_{N \times 17 \times 11 \times 11 \times 2}} \mathcal{K}_{N \times (17 \times C_3) \times 11 \times 11}, \tag{6}$$

where the bi-linear interpolation is used due to the sub-pixel based locations in the mesh and for better feature alignment.

To facilitate the structural information flow between different latent codes of the keypoints of a pose instance, we propose a simple convolutional message passing (CMP) module with three layers of Conv+BN+ReLU operations,

$$\mathcal{K}_{N \times (17 \times C_3) \times 11 \times 11} \Rightarrow [\xrightarrow[C_{in} \times 3 \times 3 \times C_{out}]{Conv+BN+ReLU}]_{\times 3} \Rightarrow \cdot \xrightarrow[C_6 \times 1 \times 1 \times 17]{Conv} K_{N \times 17 \times 11 \times 11}, \tag{7}$$

where $C_{in} \in \{(17 \times C_3), C_4, C_5\}$ and $C_{out} \in \{C_4, C_5, C_6\}$ (e.g., $C_4 = 512, C_5 = 256, C_6 = 128$ in our experiments). The resulting $K_{N \times 17 \times 11 \times 11}$ can be interpreted as keypoint attraction maps (KAMs) which are "re-focused" based on the KEMs by the CMP. To account for the specificity of different pose instances in the CMP, we adopt the Attention Normalization [17] to replace the BN in the second Conv+BN+ReLU layer, which further improves the performance in our experiments.

Through the CMP, we obtain the dynamic (a.k.a., data-driven) kernels for the 17 keypoints in a pose instance-sensitive way, which are used to refine the global heatmaps $\mathcal{H}^{\uparrow}$ for the 17 keypoints.

**iv) A local-global contextual adaptation module.** We first compute another geometric mesh with enlarged mesh window $a \times a$ (e.g., $a = 97$) for each keypoint of the $N$ pose instances, and the entire mesh is denoted by $\mathcal{M}^{L}_{N \times 17 \times a \times a \times 2}$, as done in Eqn. 5. The mesh can be interpreted as the global KEM. It is then instantiated with appearance features extracted from the global heatmaps $\mathcal{H}^{\uparrow}_{(1:17)}$, similar to Eqn. 6, and we have,

$$\mathcal{H}^{\uparrow}_{(1:17)} \xrightarrow[\text{bi-linear}]{\mathcal{M}^{L}_{N \times 17 \times a \times a \times 2}} \mathbb{H}_{N \times 17 \times a \times a} \xrightarrow[\text{reweighing}]{\mathcal{G}_{a \times a}(0, \sigma)} \bar{\mathbb{H}}_{N \times 17 \times a \times a}. \tag{8}$$

where to encode the Gaussian prior of keypoint heatmaps, the resulting pose-guided heatmaps $\mathbb{H}$ is reweighed by a Gaussian kernel $\mathcal{G}_{a \times a}(0, \sigma = \frac{a-1}{2 \times 3})$ (e.g., $\sigma = 16$ when $a = 97$) in an element-wise way. By doing so, it means that the enlarged mesh follows the $3\sigma$ principle.

Then, we apply the learned keypoint $11 \times 11$ kernels $K_{n,i}$'s (Eqn. 7) to convolve the reweighed $a \times a$ heatmap $\bar{\mathbb{H}}_{n,i}$ (Eqn. 8) in a pose instance-sensitive and keypoint-specific way, leading to **LO**cal-**GlO**bal **C**ontextual **A**daptation,

$$\bar{\mathbb{H}}_{N \times 17 \times a \times a} \xrightarrow[\text{LOGO-CA}]{K_{N \times 17 \times 11 \times 11}} \tilde{\mathbb{H}}_{N \times 17 \times a \times a}, \tag{9}$$

which represents the refined heatmaps for the 17 human pose keypoints.

**The Pose Estimation Output.** With the local-global contextually adapted heatmaps $\tilde{\mathbb{H}}_{N \times 17 \times a \times a}$, we maintain the top-2 locations for each keypoint within the $a \times a$ heatmap, and then utilize a convex average of the top-2 locations as the final predicted offset vectors (i.e. $(\Delta x'_i, \Delta y'_i)$'s in Fig. 3), and of their confidence scores as the prediction score, with a predefined weight $\lambda$ for the top-1 location (0.75 in our experiments). Together with the predicted keypoints centers $\mathcal{C}_{N \times 3}$ (Eqn. 5), the final prediction score for each keypoint is the product between the convex average confidence score and the center confidence score. We keep the keypoints whose final scores are greater than $0$. We have,

$$\{\mathcal{C}_{N \times 3}, \tilde{\mathbb{H}}_{N \times 17 \times a \times a}\} \xrightarrow[\text{Score thresholding}]{\text{Output}} \{\hat{L}^{n}_{I}; n = 1, \cdots N'\}, \tag{10}$$

where $N'$ is the number of the final predicted pose instances in an image $I$.

### 3.2.2 Loss Functions in Training

In the fully end-to-end training, we need to define loss functions for the global heatmap $\mathcal{H}$ (Eqn. 3), the refined local heatmap $\tilde{\mathbb{H}}$ (Eqn. 9), the offset field $\mathcal{O}$ (Eqn. 4), and the keypoint kernels (Eqn. 7).

**The Heatmap Loss.** The widely adopted mean squared error (MSE) loss is used. Denoted by $\mathcal{H}_{18\times h\times w}^{GT}$ the ground truth heatmaps in which each keypoint (including the center) is modeled by a 2-D Gaussian with dataset-provided mean and variance. Let $\mathbf{p}=(i,\mathbf{x})$ be the index of the domain $D$ of dimensions $18\times h\times w$. For the predicted heatmaps $\mathcal{H}_{18\times h\times w}$, the MSE loss is defined by,

$$\mathcal{L}_{\mathcal{H}} = 1/|D| \cdot \sum_{\mathbf{p}\in D} \|w(\mathbf{x})(\mathcal{H}(\mathbf{p}) - \hat{\mathcal{H}}(\mathbf{p}))\|_2^2, \tag{11}$$

where $w(\mathbf{x})$ represents the weight for the foreground and the background pixels. The foreground mask is provided by the dataset annotation. In our experiment, we set $w(\mathbf{x}) = 1$ for a foreground pixel and $w(\mathbf{x}) = 0.1$ for a background pixel.

In defining the loss function $\mathcal{L}_{\tilde{\mathbb{H}}}$ for the refined local heatmap $\tilde{\mathbb{H}}$ (Eqn. 9), the ground-truth heatmap $\tilde{\mathbb{H}}^{GT}$ is generated on-the-fly based on the mesh $\mathcal{M}_{N\times 17\times a\times a}^{L}$ (Eqn. 8) and the ground-truth keypoints using a Gaussian model with mean being the displacement between the current predicted keypoints and the ground-truth ones, and variance $\sigma$ (i.e., the standard deviation of the reweighing Gaussion prior model in Eqn. 8).

**The Offset Field Loss.** The widely adopted SmoothL1 loss [] is used. Let $\mathcal{O}_{34\times h\times w}^{GT}$ be the ground-truth offset field, and $\mathcal{C}^{GT}$ be the non-empty set of ground-truth keypoints centers (Eqn. 1). For the predicted offset field $\mathcal{O}_{34\times h\times w}$ (Eqn. 4), we have,

$$\mathcal{L}_{\mathcal{O}} = 1/|\mathcal{C}^{GT}| \cdot \sum_{\mathbf{p}\in\mathcal{C}^{GT}} \mathcal{A}(\mathbf{p}) \cdot \text{SmoothL1}\left(\mathcal{O}(\cdot,\mathbf{p}), \mathcal{O}^{GT}(\cdot,\mathbf{p}); \beta\right), \tag{12}$$

where $\mathcal{A}(\mathbf{p})$ is the area of the person centered at the pixel $\mathbf{p}$, and $\beta$ the cutting-off threshold (e.g., $\frac{1}{9}$ in our experiments), and $\text{SmoothL1}(a,b;\beta) = 0.5\times|a-b|^2/\beta$ if $|a-b|\le\beta$, otherwise $|a-b|-0.5\times\beta$.

**The OKS Loss for the Keyoint Kernels.** Consider a single predicted pose instance, learning the keypoint kernels, $K_{17\times 11\times 11}$ (Eqn. 7) is the key to facilitate the local-global contextual adaptation. To that end, the figure of merits of the KEF, $\mathcal{M}_{17\times 11\times 11\times 2}$ (Eqn. 5) needs to directly reflect the task loss, i.e., the OKS loss (Eqn. 2). With respect to the $N^{GT}$ ground-truth pose instances in an image, we can compute the similarity score per keypoint candidate in the KEF, and obtain the score tensor $S_{17\times 11\times 11\times N^{GT}}$. The score tensor is further clamped with a threshold 0.5, i.e., $S_{17\times 11\times 11\times N^{GT}} = \max(S_{17\times 11\times 11\times N^{GT}}, 0.5)$. A mean reduction is applied to the first three dimensions of the clamped score tensor to compute the matching score for each of the $N^{GT}$ pose instance. Then, the best ground-truth pose instance indexed by $n^*$ is selected in terms of the matching score, and its matching score is denoted by $s_{n^*}$. Based on the selected ground-truth pose instance, we compute the per-keypoint similarity score for the predicted pose instance at hand, denoted by $s_k$ ($k\in[1,17]$). Then, the loss function fo the keypoint kernels are defined by,

$$\mathcal{L}_K = s_{n^*} \cdot \sum_{k,i,j} s_k \cdot |K_{k,i,j} - S_{k,i,j,n^*}|^2. \tag{13}$$

**The Total Loss** is then defined by $\mathcal{L} = \mathcal{L}_{\mathcal{H}} + \mathcal{L}_{\tilde{\mathbb{H}}} + \lambda\cdot(\mathcal{L}_{\mathcal{O}} + \mathcal{L}_K)$, where the trade-off parameter $\lambda$ is used to balance the different loss items ($\lambda = 0.01$ in our experiments).

## 4 Experiments

In this section, we present detailed experimental results and analyses of the proposed LOGO-CAP. **Our PyTorch source code will be released for reproducibility.**

**Datasets.** We use two datasets in our experiments: **The COCO dataset [18]** is the most popular testbed for human pose estimation. It consists of 65k, 5k and 20k images with human pose well-annotated in the training, validation and testing datasets respectively. In all experiments, the proposed LOGO-CAP is trained using the 65k training images. **The OCHuman dataset** [34] is one popular *testing-only* dataset for evaluating human pose estimation under the occlusion scenarios. It consists of a total number of 4713 images with 8110 detailed annotated human pose instances using the COCO keypoint configuration. All the annotated 8110 human pose instances have occlusions with the maxIOU$\ge$ 0.5. Furthermore, 32% instances are more challenging with the maxIOU$\ge$ 0.75.

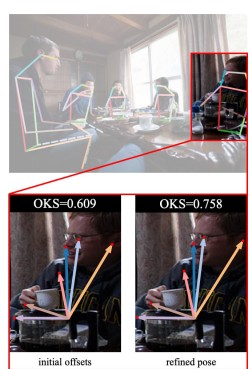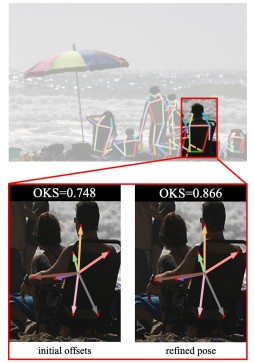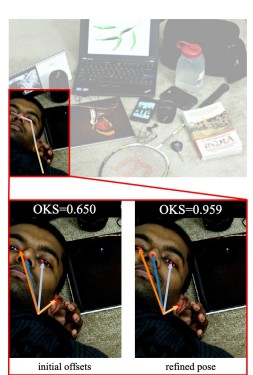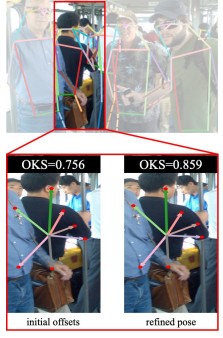

Figure 4: Examples of human pose estimation in the COCO val-2017 dataset by the proposed LOGO-CAP with the HRNet-W32 backbone. *Top:* The COCO skeleton template based visualization. *Bottom*: The close-up visualization and OKS comparisons between the initial center-offset estimation and the refined keypoints.

Table 2: Evaluation results on the `COCO-val-2017` and `COCO-testdev-2017` dataset. For HGG [13] and SimplePose [16], the multi-scale inference$^\dagger$ is applied on the testdev-2017 dataset. For DEKR [11] that uses an rescoring network to get the final predictions, we report both the performance with and without rescoring (which is the fair baseline for our LOGO-CAP). The numbers of SPM [21] and HGG [13] are extracted from their papers.

| | Method | Backbone | COCO-val-2017 | | | | | COCO-testdev-2017 | | | | |
|---|---|---|---|---|---|---|---|---|---|---|---|---|
| | | | AP [%] | AP$^{50}$ [%] | AP$^{75}$ [%] | AP$^M$ [%] | AP$^L$ [%] | AP [%] | AP$^{50}$ [%] | AP$^{75}$ [%] | AP$^M$ [%] | AP$^L$ [%] |
| Grouping | OpenPose [35] | VGG-19 | 61.0 | 84.9 | 67.5 | 56.3 | 69.3 | 61.8 | 84.9 | 67.5 | 57.1 | 68.2 |
| | PifPaf [15] | ResNet-152 | 67.4 | 86.9 | 73.8 | 63.1 | 74.1 | 66.7 | 87.8 | 73.6 | 62.4 | 72.9 |
| | PersonLab [22] | ResNet-152 | 66.5 | 86.2 | 71.9 | 62.3 | 73.2 | 66.5 | 88.0 | 72.6 | 62.4 | 72.3 |
| | AE [20, 5] | HrHRNet-W32 | 67.1 | 86.2 | 73.0 | 61.5 | 76.1 | 66.4 | 87.5 | 72.8 | 61.2 | 74.2 |
| | | HrHRNet-W48 | 69.9 | 87.2 | 76.1 | 65.4 | 76.4 | 68.4 | 88.2 | 75.1 | 64.4 | 74.2 |
| | HGG [13] | Hourglass | 60.4 | 83.0 | 66.2 | – | – | 67.6$^\dagger$ | 85.1$^\dagger$ | 73.7$^\dagger$ | 62.7$^\dagger$ | 74.6$^\dagger$ |
| | SimplePose [16] | IMHN | 66.1 | 85.9 | 71.6 | 59.8 | 76.2 | 68.5$^\dagger$ | 86.7$^\dagger$ | 74.9$^\dagger$ | 66.4$^\dagger$ | 71.9$^\dagger$ |
| Direct | SPM [21] | Hourglass | – | – | – | – | – | 66.9 | 88.5 | 72.9 | 62.6 | 0.731 |
| | CenterNet [35] | Hourglass | 64.0 | 85.6 | 70.2 | 59.4 | 72.1 | 63.0 | 86.8 | 69.6 | 58.9 | 70.4 |
| | DEKR [11] (w. Rescoring) | HRNet-W32 | 68.0 | 86.7 | 74.5 | 62.1 | 77.7 | 67.3 | 87.9 | 74.1 | 61.5 | 76.1 |
| | | HRNet-W48 | 71.0 | 88.3 | 77.4 | 66.7 | 78.5 | 70.0 | 89.4 | 77.3 | 65.7 | 76.9 |
| | DEKR [11] (w.o. Rescoring) | HRNet-W32 | 67.2 | 86.3 | 73.8 | 61.7 | 77.1 | 66.6 | 87.6 | 73.5 | 61.2 | 75.6 |
| | | HRNet-W48 | 70.3 | 87.9 | 76.8 | 66.3 | 78.0 | 69.3 | 89.1 | 76.7 | 65.3 | 76.4 |
| | LOGO-CAP (Ours) | HRNet-W32 | 69.6 | 87.5 | 75.9 | 64.1 | 78.0 | 68.2 | 88.7 | 74.9 | 62.8 | 76.0 |
| | | HRNet-W48 | **72.2** | **88.9** | **78.9** | **68.1** | **78.9** | **70.8** | **89.7** | **77.8** | **66.7** | **77.0** |

## 4.1 Results on the COCO dataset

Fig. 4 shows some qualitative examples of human pose estimation by the proposed LOGO-CAP. More examples will be provided in the supplementary material.

The proposed LOGO-CAP is compared with prior arts including OpenPose [3], PifPaf [15], PersonLab [22], AE [20] and DEKR [11]. As reported in Table 2, the proposed LOGO-CAP outperforms all of them on both both validation and test-dev datasets.

In comparisons to the best-performing grouping approach, AE [20] with a larger backbone HrHRNet-W48 [5], our LOGO-CAP obtains competitive performance with a smaller HRNet-32 backbone, and improves the AP score with HRNet-W48 backbone on the validation and testdev datasets by `2.3` and `2.5` points, respectively. For the fully differentiable grouping approach HGG [13], our LOGO-CAP achieves better performance by a significantly large margin, more than `9.2` points on the validation set under the single-scale testing. Although the performance of HGG is improved by the multi-scale testing on the test-dev set, the performance of our LOGO-CAP is still significantly better without using the multi-scale testing scheme.

In comparisons to the direct regression based approaches, our LOGO-CAP obtains the *best results* without incurring either the matching scheme used in CenterNet [35] or the additional rescoring network used in DEKR [11]. When we disable the rescoring network for DEKR [11] for fair comparisons, our LOGO-CAP significantly improves the AP on the validation and testdev datasets by `2.4` points and `1.6` points respectively when HRNet-W32 is used as backbone. The larger backbone is beneficial for both DEKR and our method, which further improves the AP score of our LOGO-CAP to `72.2` and `70.8` on the validation and test-dev dataset respectively, outperforming DEKR by `1.9` and `1.5` respectively.

Table 3: Results on the OCHuman validation and testing datasets [34].

| | Methods | Backbone | Val. AP [%] | Test AP [%] |
|---|---|---|---|---|
| Top-down | RMPE [7] | Hourglass | 38.8 | 30.7 |
| | SBL [32] | ResNet-50 | 37.8 | 30.4 |
| | SBL [32] | ResNet-152 | 41.0 | 33.3 |
| Bottom-up | AE [20] | Hourglass | 32.1 | 29.5 |
| | HGG [20] | Hourglass | 35.6 | 34.8 |
| | DEKR [11] | HRNet-W32 | 37.9 | 36.5 |
| | | HRNet-W48 | 38.8 | 38.2 |
| | LOGO-CAP | HRNet-W32 | 39.0 | 38.1 |
| | (Ours) | HRNet-W48 | **41.2** | **40.4** |

Table 4: The single image inference speed comparison for bottom-up human pose estimation approaches.

| Method | AP [%] | Backbone | Time ↓ [ms] | FPS ↑ |
|---|---|---|---|---|
| PifPaf [15] | 67.4 | ResNet-152 | 213 | 4.68 |
| AE [20, 5] | 67.1 | HrHRNet-W32 | 560 | 1.78 |
| CenterNet [35] | 64.0 | Hourglass | 147 | 6.80 |
| DEKR [11] | 68.0 | HRNet-W32 | 63 | 15.8 |
| DEKR [11] | 71.0 | HRNet-W48 | 139 | 7.21 |
| LOGO-CAP | 69.6 | HRNet-W32 | 48 | **20.7** |
| LOGO-CAP | 72.2 | HRNet-W48 | 112 | 8.95 |

## 4.2 Results on the OCHuman dataset

Table 3 shows that our LOGO-CAP achieves the best AP performance on both the validation and testing datasets by significant margins of 2.4 and 2.2 points in comparing with the bottom-up approaches. For the top-down approaches, although they obtain strong AP scores on the validation split, there exists a large performance gap between the validation and testing sets. In comparisons to DEKR [11] (with the rescoring network), our LOGO-CAP improves the performance from 37.9 to 39.0 and from 36.5 to 38.1 on the validation and testing splits with the same backbone HRNet-W32, respectively. The similar improvement is observed when the HRNet-W48 backbone is used, outperforming both bottom-up and top-down approaches.

## 4.3 Inference Speed

In comparing the inference speed, we test all the models on a single TITAN RTX GPU for its popularity in practice. The average inference speed, FPS (frames per second), over the 5000 images in COCO-val-2017 is used for the comparison. For DEKR [11], we re-implement their inference code with better speed obtained for fair comparisons at the algorithm level. For methods that have post-processing schema on CPU, only one thread is used. As shown in Table 4, our LOGO-CAP runs significantly faster than PifPaf [15] and AE [20]. The CenterNet [35] runs slower than DEKR and our LOCO-CAP as it requires a post-processing scheme to match the predicted offsets to the keypoints obtained from heatmaps. Comparing with DEKR, the speed improvement of our LOGO-CAP is from the lightweight design of head modules since the same backbones are used. For the comparisons in Table 2, we run the models with different resolutions of testing images.

## 4.4 Potentials and Limitations of the Proposed LOGO-CAP

Consider the generic applicability of the center-offset formulation to many computer vision tasks as demonstrated in [35], we hypothesize that the proposed LOGO-CAP has a great potential to remedy the lack of sufficient accuracy using the vanilla center-offset method in those tasks. We also notice that the minimally-simple design in learning the "Slow Keypointer" can be relaxed for different accuracy-speed trade-offs in practice. For example, for the convolutional message passing module, an alternative method could be the Transformer model [31], which potentially will further improve the performance at the expense of inference speed. We leave these for future work.

## 5 Conclusion

This paper focuses on deep learning based formulation for bottom-up human pose estimation. It presents a method of learning LOcal-GlObal Contextual Adaptation for Pose estimation, dubbed as LOGO-CAP. The proposed LOGO-CAP is built on the conceptually simple center-offset paradigm and addresses its drawback of lacking the capability of accurately localizing human pose keypoints. The key idea of our LOG-CAP is to lift the center-offset predicted keypoints to keypoint expansion maps (KEMs),which counters the inaccuracy and uncertainty of the initial keypoints. Two types of KEMs are introduced in two parallel modules on top of the feature backbone. Local KEMs are used to learn keypoint attraction maps (KAMs) via a convolutional message passing module that accounts for the structured output prediction nature of human pose estimation. Global KEMs are used to learn local-global contextual adaptation which convolves global KEMs using the KAMs as kernels. The refined global KEMs are used in computing the final human pose estimation. The proposed LOGO-CAP obtains state-of-the-art performance in COCO val-2017 and test-dev 2017 datasets for bottom-up human pose estimation. It also achieves state-of-the-art transferability performance in the OCHuman dataset with the COCO trained models.

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
