# Learning Local-Global Contextual Adaptation for Fully End-to-End Bottom-Up Human Pose Estimation

### Supplementary Material

## 1 Experimental Settings

Our PyTorch source code will be released. We briefly present the details of training and testing as follows.

**Training.** We train two LOGO-CAP networks with the ImageNet pretrained HRNet-W32 and HRNet-W48 [9] as the feature backbone respectively on the COCO-train-2017 dataset [6]. Common training specifications are used for simplicity in experiments. The Adam optimizer [4] is used with default coefficients $\beta_1 = 0.9$ and $\beta_2 = 0.999$. For both the backbones, the total number of epochs is set to $140$ and the batch size is set to $12$ images per GPU card. The same learning rate schedule is used for both models. The learning rate is initially set to $0.001$ and then decayed to $10^{-4}$ and $10^{-5}$ at the 90-th and 120-th epoch respectively. We use $4$ and $8$ V100 GPUs to accelerate the training for the two LOGO-CAP models with HRNet-W32 and HRNet-W48 respectively. The resolution of training images is $512 \times 512$ and $640 \times 640$ for the two models respectively. Following the widely adopted experimental settings in [10], the data augmentations in training include (1) random rotation with the rotation degree from $-30°$ to $30°$, (2) random scaling with the factor in the range of $[0.75, 1.5]$, (3) random translation in the range $[-40\text{pix}, 40\text{pix}]$ along both $x$ and $y$ directions, and (4) random horizontal flipping with the probability of $0.5$.

Similarly, for different hyperparamters such as the trade-off parameter $\lambda$ in the total loss (Section 3.2.2 in the submission), we did not run computationally expensive hyperparameter optimization for simplicity.

**Testing.** We focus on the single-scale testing protocol in the COCO keypoint benchmark for the sake of efficient human pose estimation. In the testing phase, the short side of input images is resized to a specific length (*e.g.*, 384, 512, or 640 pixels) and keep unchanged the aspect ratio between the height and the width. As commonly adopted in many bottom-up pose estimation approaches (*e.g.*, AE [7], HrHRNet [1], DEKR [2]), the flip testing is used as our default setting for the fair comparison. In the implementation, we feed the stacked tensor with an input image and a horizontally-flipped one together to get the global heatmaps and the offset fields. The flipped outputs are then averaged (according to the flip index) to get the final global heatmaps and the offset fields. For the computation of local heatmaps and the local-global adaptation, only the non-flipped outputs are used for the final predictions.

## 2 The Computation of the Empirical Upper Bound

We elaborate on the details of computing the empirical upper bound of performance for a vanilla center-offset pose estimation method (Table.1 in the submission).

**Network Architecture.** The vanilla center-offset regression baseline uses the ImageNet pretrained HRNet-W32 [9] as the backbone, and the same modules as in our LOGO-CAP+HRNet-W32 for the center heatmap regression and the offset vector regression. See Fig.3 and Section 3 in the submission for detailed specifications. We present the details of computing keypoint expansion maps (KEMs) that are used in calculating the empirical uppper bound as follows.

**Computation of Keypoint Expansion Maps.** Denoted by $\mathcal{P}_{N \times 17 \times 2}$ the initial pose parameters (i.e., the 2-D locations for the 17 keypoints of the $N$ pose instances) estimated by the vanilla center-offset method, we expand each of the estimated keypoints with a local $11 \times 11$ mesh grid, that is to lift a keypoint to a 2-D mesh to counter the estimation uncertainty. As shown in the Alg. 1, we use the COCO benchmark provided keypoint sigmas to scale the unit length of the meshgrid for different types (*e.g.*, nose, eyes, hips) of keypoints. After getting the expanded keypoint meshes $\mathcal{M}$ of the initial poses, we compute their keypoint similarities $\mathcal{S}_{N \times 11 \times 11 \times K \times 17}$ between the groundtruth keypoints $\mathcal{G}_{K \times 17 \times 3}$ and the keypoint expansion maps. After applying the sum reduction on the similarity tensor $\mathcal{S}_{N \times 11 \times 11 \times K \times 17}$ along the 2-nd, 3-rd and the last axes, we have known the optimal correspondence (including the low-quality matches) for each center anchor, denoted by $\mathcal{S}_{N \times 11 \times 11 \times 17}$. Then, the pose with the maximal similarity in the $11 \times 11$ local window for each center anchor are used as the best one to compute the empirical upper bound on the fully-annotated COCO-val-2017 dataset.

---

**Algorithm 1:** Computation of Keypoint Expansion Maps in a PyTorch-like style

```
1  coco_sigmas = torch.tensor([0.026, 0.025,0.025, 0.035, 0.035, 0.079, 0.079, 0.072, 0.072,
     0.062, 0.062, 0.107, 0.107, 0.087, 0.087, 0.089, 0.089 ]) #Keypoint sigmas for the
     COCO dataset.
2  def KptsExpansionCoco(𝒫,ksize=11)
      #Initial poses 𝒫:  Nx17x2
3  |   radius = ksize // 2
4  |   dy, dx = torch.meshgrid(torch.arange(-radius,radius),torch.arange(-radius,radius))
      #dx, dy:  ksize×ksize
5  |   dy, dx = dy.reshape(1, 1, ksize, ksize), dx.reshape(1, 1, ksize, ksize)
6  |   scale = coco_sigmas.reshape(1, 17, 1, 1)/coco_sigmas.min() #using different
      expansion rate for the different keypoint categories.
7  |   dy = dy * scale #dy:  1x17x11x11
8  |   dx = dx * scale #dx:  1x17x11x11
9  |   dxy = torch.stack((dx,dy),dim=-1) #dxy:  1x17x11x11x2
10 |   𝓜 = 𝒫.reshape(N, 17, 1, 1, 2) + dxy #𝓜:  Nx17x11x11x2
11 |   return 𝓜
```

---

# 3 The Speed-Accuracy Comparisons

We elaborate on the speed-accuracy comparisons in Fig. 2 in the submission, which is reproduced in Fig. 1.

We compare our LOGO-CAP with state-of-the-art bottom-up pose estimation approaches in terms of the speed-accuracy trade-off using different input resolutions of testing images.

- Both the proposed LOGO-CAP method and the DEKR [2] method have two versions of models trained with the HRNet-W32 backbone at the resolution of $512 \times 512$, and the HRNet-W48 backbone at the resolution of $640 \times 640$, respectively. The two versions of models are denoted by $W32$ and $W48$ respectively in Fig. 1. In comparing the speed-accuracy trade-offs, we evaluate each of the two versions of models with two different resolutions: For $W32$, we test the models, LOGO-CAP and DEKR, under the resolutions of 384 and 512 for short side of the testing images, which are denoted by $W32 - 384$ and $W32 - 512$ in Fig. 1 respectively. Similarly, it is done for the $W48$ models. The longer side of the testing images are computed according to their original aspect ratios.

- For the CenterNet method [11], we report the result obtained using the large Hourglass backbone [8] due to its better AP score. The testing resolution is $512$ in the short side of the images.

- For the PifPaf method [5], we report their results using three different backbones of ResNet-50, ResNet-101 and ResNet-152 [3]. We follow the image resolution setting presented in the original paper [5] that resizes the long side of the testing images to be $641$ and keep the original aspect ratio.

- For the associative embedding approach [7], the result using the Higher-HRNet-W32 [1] as the backbone is reported for the concern of inference speed. The testing resolution is also $512$ in the short side of the testing image.

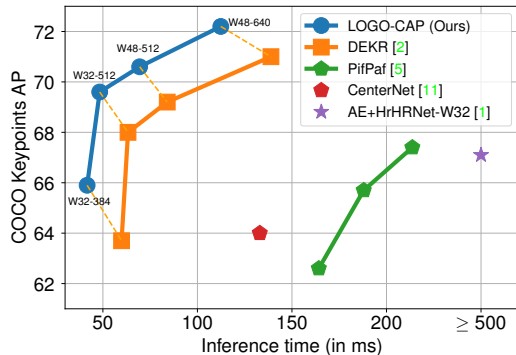

Figure 1: Speed-accuracy comparisons between our LOGO-CAP and prior arts on the COCO val-2017 dataset. W$x$-$Y$ (e.g. W32-384) means that a model uses the backbone HRNet-W$x$ (HRNet-W32) and is tested with the image resolution $Y$ in the short side.

As shown in Fig. 1, our approach obtains the best speed-accuracy trade-off with different backbones and image resolutions. When using HRNet-W32 as the backbone, our LOGO-CAP uses the low-resolution images (with 384 pixels in the short side) obtains `the AP of 65.9` while approaching real-time performance with `the FPS of 24.0`, which surpasses DEKR [2] by `2.2 AP` and `7.3 FPS`. Benefitting from our design rationales of simplicity and fully end-to-end learning, our approach also obtains the best performance in both aspects of speed and accuracy when increasing the image resolution to 512 and 640.

## 4 More Qualitative Results

**Results on the COCO-val-2017 and the OCHuman Datasets.** Fig. 2 shows examples of pose estimation in the two datasets by the proposed LOGO-CAP with the HRNet-W32 backbone. Our proposed LOCO-CAP is able to handle large structural and appearance variations in human pose estimation.

**Fast pose estimation for video frames.** To justify the potential of our proposed approach in practical applications, we run our LOGO-CAP (W32 model) on two videos that have the resolution of $1280 \times 720$ from YouTube. We follow our testing protocol to resize the short side of the video frames to 512 pixels and keep their original aspect ratios for inference. Without using any pose tracking techniques, our LOGO-CAP achieves fast and accurate human pose estimation. Please click the following anonymous google drive links for the demo videos:

- https://bit.ly/3z2t8fA (video credit: https://youtu.be/2DiQUX11YaY)

- https://bit.ly/3cghWT4 (video credit: https://youtu.be/kTvzU1sGSyA)

In these two demo videos, the instantaneous FPS for each video frame is marked in the left corner of the video.

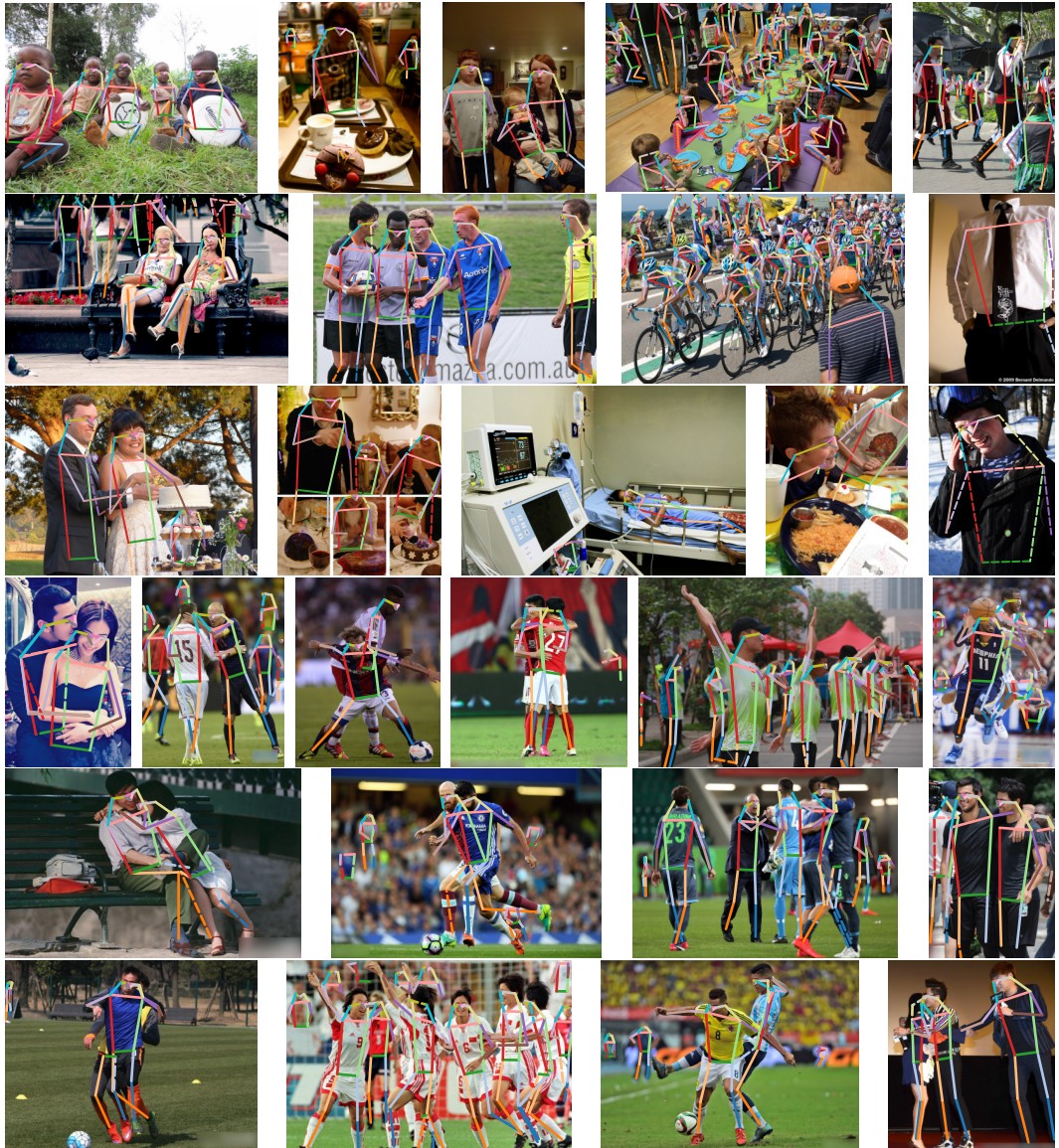

Figure 2: Qualitative results of our LOGO-CAP (HRNet-W32). All images were picked thematically without considering our algorithms performance. The first three rows display our approach on the COCO-val-2017 dataset and the last three ones show our results on the OCHuman test dataset.