# OpenReview forum: "Learning Local-Global Contextual Adaptation for Fully End-to-End Bottom-Up Human Pose Estimation"
_NeurIPS.cc/2021/Conference — NeurIPS 2021 Submitted_

### Official Review · Reviewer_1GBc · 2021-07-05

**Rating:** 8
**Confidence:** 4

**Summary:**

This paper proposes a center-offset based bottom-up human pose estimation architecture. The proposed method first estimates the centers and offsets of keypoints to get the initial poses. Then for each keypoint, estimate the parameters of an 11x11 convolution kernel in an instance-aware way. Finally, for each keypoint, a larger window is extracted from the global heatmap, and the learned 11x11 kernels are applied to convolve the extracted heatmap.


**Limitations And Societal Impact:**

Since each keypoint of each person has to be dealt with separately, the speed of the method proposed in this paper should be greatly affected by the number of people in an image, and the memory usage should also be large, but this paper does not mention it. If so, the author can add some relevant descriptions to the paper. If not, the author can mention it in the paper and leave it for follow-up work.


**Main Review:**

The method proposed by this paper is novel and technically sound.
The idea of combining keypoint features that belong to the same person to obtain the convolution kernel of a single keypoint is interesting. Extracting a 97x97 pose-guided heatmap for each keypoint introduces high resolution to the bottom-up architecture, what is exactly the bottom-up architecture needed.
Of course, this paper also has some problems. First, the "Fast Keypointer" and "Slow Keypointer" mentioned at the beginning of this paper do not actually relate to speed. The "Slow Keypointer" proposed is actually a refinement of the coarse estimated pose. This statement affects readers' understanding to a certain extent. Second, there are many tricks used in this paper, such as Attention Normalization, reweigh heatmaps using gaussian kernel and OKS loss. But ablation study has not been carried out in the experiment to show the influence of these tricks. Adding an experiment like this would be more convincing. Last, since each keypoint of each person has to be dealt with separately, the speed of the method proposed in this paper should be greatly affected by the number of people in an image, and the memory usage should also be large, but this paper does not mention it. I don’t know if some techniques are used to alleviate these two problems.
All in all, the conclusions of this paper are significant. After DEKR, it proves again that the center-offset architecture can perform high-precision pose estimation.


**Time Spent Reviewing:**

6

---

> ### Author Response · Authors · 2021-08-10
> **Reply**
>
> **Before addressing your valuable comments, we appreciate you for the recognition of the proposed work: "All in all, the conclusions of this paper are significant. After DEKR, it proves again that the center-offset architecture can perform high-precision pose estimation." We also would like to highlight that as shown by the surprisingly strong observations in the Table 1 of the submission, the center-offset architecture worth more studies towards highly efficient yet accurate human pose estimation.**
>
>
>
> # Detailed Responses:
>
> ### 1. The "Fast Keypointer" and "Slow Keypointer" mentioned at the beginning of this paper do not actually relate to speed. The "Slow Keypointer" proposed is actually a refinement of the coarse estimated pose.
>
> **Ans**:  We are sorry for the confusion. As stated in L4-6 and L55-L56, we introduce our proposed LOGO-CAP by analogy to the modes of thought suggested by Daniel Kahneman in the book of  *"Thinking, Fast and Slow"*. It would be indeed confusing for the readers who are not familiar with this book. We will revise it to make it clearer.
>
> ### 2. There are many tricks used in this paper, such as Attention Normalization, re-weighting heatmaps using Gaussian kernel and OKS loss. But ablation study has not been carried out in the experiment to show the influence of these tricks. Adding an experiment like this would be more convincing.
>
>
> **Ans**: Thank you for your suggestions to make our paper stronger. As shown in the Table 1 below, we conduct ablation studies on the three components: the OKS loss, the Gaussian reweighing method for heatmaps and the Attentive Normalization. Compared with the baseline without the three components, both (a) using the OKS loss only and (b) using the reweighing method only can dramatically improves the final performance. As shown in (d) and (e), the Attentive Normalization does not bring any improvement without jointly training the three components together. For the joint learning, the Attentive Normalization brings an AP improvement by 0.6 AP, comparing (c) and (f).  When comparing with the best-performing approach DEKR [11], our LOGO-CAP still outperforms it by 1.0 AP (i.e., from 68.0 to 69.0) without using Attentive Norm [17]. We will add these in revision.
>
> **Table 1. Ablation studies on the three components: the OKS loss, the Gaussian reweighing method for heatmaps and the Attentive Normalization.**
>
> | HRNet-W32 | OKS Loss | Reweigh | AttNorm | AP  | AP$^{50}$ | AP$^{75}$ | AP$^M$ | AP$^L$ |
> | :-------: | :------: | :-----: | :-----: | :--: | :----: | :----: | :---: | :---: |
> | baseline  |    -     |    -    |    -    | 60.0 |  84.4  |  66.4  | 54.0  | 71.1  |
> |    (a)    |    ✔     |    -    |    -    | 66.1 |  86.7  |  72.7  | 60.0  | 75.6  |
> |    (b)    |    -     |    ✔    |    -    | 67.6 |  87.0  |  74.3  | 62.1  | 76.7  |
> |    (c)    |    ✔     |    ✔    |    -    | 69.0 |  87.0  |  75.2  | 63.4  | 77.5  |
> |    (d)    |    ✔     |    -    |    ✔    | 65.8 |  86.8  |  72.3  | 59.3  | 75.4  |
> |    (e)    |    -     |    ✔    |    ✔    | 67.5 |  86.6  |  74.1  | 62.2  | 76.7  |
> |    (f)    |    ✔     |    ✔    |    ✔    | 69.6 |  87.5  |  75.9  | 64.1  | 78.0  |
> |
>
>
> The ablation studies verify the effectiveness of integrating the three components in the proposed method. In addition to the empirical performance justification, we also would like to point out the underlying intuition and necessity for the three components: (i) The OKS loss. Human pose estimation is a classic structured output prediction problem. On the one hand, before the recent resurgence of DNNs, developing structural loss functions is one most important problem in the structured output problem formulation (e.g., the classic structural SVMs). Structural loss functions are typically overlooked in DNN based human pose estimation methods for simplicity. On the other hand, the OKS metric is one of the most widely used one in different benchmarks. For the proposed method, integrating the OKS loss in training bridges the discrepancy between training and testing in many SOTA methods such as those compared in this paper.  (ii) The Gaussian reweighing method. It is commonly used in almost all heatmap-based keypoint regression methods, accounting for the intrinsic uncertainty/ambiguity of keypoints in discretized image lattice. (iii) The Attentive Normalization. It is introduced to account for the underlying structural and appearance variations of different keypoints in the convolutional messaging passing computation in a light-weight way. As mentioned in the submission, more sophisticated and computationally expensive methods such as the Transformer models could also be exploited. In sum, (i) and (ii) are entailed due to the nature of the human pose estimation problem and the heatmap based regression formulation, while (iii) is used by considering the trade-off between efficiency and accuracy.
>
> So, on a slide note, the three components may not be necessarily treated as tricks, as they are intuitively entailed for improving the representational power of the proposed method for better learning the human pose estimator that is of the nature of structured output prediction. We will add these in revision.
>
> ### 3. Last, since each keypoint of each person has to be dealt with separately, the speed of the method proposed in this paper should be greatly affected by the number of people in an image, and the memory usage should also be large, but this paper does not mention.
>
>
> **Ans**: We will add the complexity analysis of the proposed method (excluding the common feature backbone). There are two sides in the consideration:
>
>  (i) The theoretical time and memory footprint complexity are indeed of the linear relationship with the number of person instances and the number of keypoints. Denoted by $P$ the number of person instances in an image (e.g. $0\leq P \leq 30$ in a typical image), $K$ the number of keypoints for each person (e.g., $K=17$ in the COCO pose structure), $\omega$ the kernel size of the local KEMs (e.g., $\omega=11$) and $H$ the side length of the cropped square global heatmaps (e.g. $H=97$). The time complexity of applying the convolution operator is $\mathcal{O}(P\cdot K\cdot \omega^2\cdot H^2)$. We note that $\omega$ and $H$ are agnostic to the size of input images.
>  The memory footprint of the proposed local-global contextual adaption head includes three parts as follows: (1) the generation of local KEMs uses $\mathcal{O}(P\cdot K \cdot \omega^2)$ memory footprint, (2) the estimation of local KAMs uses $\mathcal{O}(P\cdot c)$ where $c$ is a constant number for the convolutional message passing layers for each person, (3) the cropped global heatmaps takes $\mathcal{O}(P\cdot K \cdot H^2)$, and (3) the computation of the convolutional adaptation takes $\mathcal{O}\left(P\cdot K \cdot (\omega^2 + H^2)\right)$. The overall memory complexity is $\mathcal{O}(P\cdot K \cdot H^2)$ as $\omega$ is greatly smaller than $H$. Both the time and memory complexity are easily manageable by a modern GPU card (e.g., Titan RTX used in our experiments).
>
> (ii) The empirical observation is that the proposed method can obtain near real-time speed (Table 4 in the submission), significantly faster than prior arts, due to the proposed light-weight implementation. The analyses are provided in the Section 4.3 in the submission with the details of comparisons provided in the Section 3 in the supplementary. The breakdown of inference times is shown in Table 2.
>
> **Table 2: The breakdown of inference time of the proposed LOGO-CAP method.  The inference time is averagelycomputed using a NVIDIA TITAN RTX GPU on the COCO-val-2017 dataset. For each model, we seperately report the averaged inference time across 5000 images, the averaged inference time in the images that detect only one person, the avaraged inference time in the images that have 30 persons.**
>
> |                | Number of Persons | Backbone | Local KEMs | Local KAMs | Global KAMs |
> | :------------: | :---------------: | :------: | :--------: | ---------- | ----------- |
> | LOGO-CAP (W32) |         -         | 38.6 ms  |  3.05 ms   | 2.49 ms    | 2.85 ms     |
> |                |         1         |          |  2.39 ms   | 1.14 ms    | 1.12 ms     |
> |                |        30         |          |  3.69 ms   | 3.49 ms    | 5.87 ms     |
> |
> | LOGO-CAP (W48) |         -         | 99.9 ms  |  4.18 ms   | 3.00 ms    | 3.34 ms     |
> |                |         1         |          |  3.19 ms   | 1.10 ms    | 1.07 ms     |
> |                |        30         |          |  2.97 ms   | 3.59 ms    | 5.97 ms     |
> |

---

> > ### Comment · Reviewer_1GBc · 2021-08-15
> > **Reply: Reply**
> >
> > Thank you for your reply.
> >
> > From my personal point of view, current top-down pose estimation methods are relatively mature and the main way to enhance performance is to improve the backbone. While bottom-up approaches rely more on design because of the problem of "how to group people". The method you proposed verifies the possibility of "center-offset" architecture in pose estimation again, and I think this will lead to the emergence of more good bottom-up methods, just like the emergence of anchor-free methods in object detection. Considering the proposed method can achieve SOTA performance and real-time speed at the same time, I think this is the best paper I've seen during this review.

---

> > > ### Author Response · Authors · 2021-08-17
> > > **Thanks!**
> > >
> > > We sincerely appreciate you for the recognition of our work!

---

### Official Review · Reviewer_k2Ew · 2021-07-13

**Rating:** 3
**Confidence:** 5

**Summary:**

This paper introduces an end-to-end human pose estimation method, termed LOGO-CAP. It proposes a local-global contextual adaptation formulation and applies KAMs and the global KEMs to refine the estimated center-offset poses. It achieves 72.2 mAP on COCO val-2017 dataset.

**Limitations And Societal Impact:**

Yes. The authors adequately described the limitations of the work.
This paper does not have obvious potential negative societal impact.

**Main Review:**

Pros:
This work proposes a novel local-global contextual adaptation formation for accurate pose estimation. Comparing with bottom-up center-offset methods, LOGO-CAP achieves a better trade-off between accuracy and efficiency, which is important in the task of human pose estimation.

Cons:
1.	This paper argues that “it is the first fully end-to-end trainable method that achieves state-of-the-art performance”, which is over-claimed. 1) There are several end-to-end methods for human pose estimation (e.g., Mask RCNN, bottom-up methods [35, 20] and bottom-up regression methods [28, 11]); 2) LOGO-CAP does not achieve the state-of-the-art performance (SimpleBaseline[32], HRNet[27]).
2.	The “Slow” keypointer, which introduces KAMs and the global KEMs to refine the poses estimated by center-offset formulation, is applied at person instance level (or even at joint level). It is more like a top-down approach rather than a bottom-up approach. It is likely that the center-offset formulation only provides the coarse position of persons or joints, then they are processed individually. Their inference time increases with the number of people. The reviewer believes that both top-down approaches and bottom-up approaches should be compared in Table2.
3.	This paper lacks of ablation study to show the effect of proposed modules. It should clearly introduce where the improvement over the baseline come from. The following experiment results should be given: a) the effect of local-global contextual adaptation; b) the effect of Gaussian reweighting; c) the effect of convolution message passing module; d) The reviewer is also curious about the result of applying the local window search (the argmax) on the KAMs, which is similar to Table.1 Emp. Bound.
4.	The inference time of the proposed method increases with the number of persons, as mentioned above. The details of how to test the inference speed for multi-person images and the reasons why LGO-CAP show superiority over the bottom up methods (DEKR[11]) on inference speed should be given.
5.	The inaccuracy of global KEMs comes from the global keypoint heatmaps, while the KAMs measure the error of center-offset results. Here is the mismatch between heatmaps and center-offsets. How can KAMs help refine global KEMs?
6.	The writing can be improved. Notations in this paper are not well defined. There are also too many similar concepts with different names, e.g. Local heatmaps, local KEMs, KAMs. It would be better if they use a consistent name throughout the paper. Some important details of the proposed approach are missing (e.g., how to compute the geometric mesh of the global KEMs).

**Time Spent Reviewing:**

6

---

> ### Author Response · Authors · 2021-08-10
> **Reply: Part 1**
>
> ### 1. This paper argues that “it is the first fully end-to-end trainable method that achieves state-of-the-art performance”, which is over-claimed. 1) There are several end-to-end methods for human pose estimation (e.g., Mask RCNN, bottom-up methods [35, 20] and bottom-up regression methods [28, 11]); 2) LOGO-CAP does not achieve the state-of-the-art performance (SimpleBaseline[32], HRNet[27]).
>
> **Ans**: With all due respect, we clearly state the contribution under the context that "The proposed LOGO-CAP makes three main contributions to the field of bottom-up human pose estimation" (L. 158-159). We are aware of all the great work mentioned in the comment. We will revise the claim to be more precise: It achieves state-of-the-art bottom-up human pose estimate performance. It is fully end-to-end trainable, unlike many other bottom-up human pose estimation methods which entail separate post-hoc processing to improve performance. Literally speaking, the statement itself is not over-claimed due to the that-clause, but indeed not as precise as needed.
>
>
> ### 2. The “Slow” keypointer, which introduces KAMs and the global KEMs to refine the poses estimated by center-offset formulation, is applied at person instance level (or even at joint level). It is more like a top-down approach rather than a bottom-up approach. It is likely that the center-offset formulation only provides the coarse position of persons or joints, then they are processed individually. Their inference time increases with the number of people. The reviewer believes that both top-down approaches and bottom-up approaches should be compared in Table2.
>
> **Ans**: Consider state-of-the-art deep learning based approaches, there may not be strict line between top-down and bottom-up human pose estimation methods. For examples, among many others which are commonly categorized as bottom-up human pose estimated methods, the DEKR method utilizes a scoring net for pose candidates, which may be seen as a top-down component. The Associative Embedding method uses a pose instance guided computing order in inference ("An ordering is determined by first considering joints around the head and torso and gradual moving out to the limbs."), which may also be treated as a top-down operation. So, we think it is fair to compare our method with the main bottom-up stream in experiments.
>
> More specifically, for state-of-the-art approaches that are categorized as top-down such as the HRNet [27] and the SimpleBaseline [32], there are two individual networks for person detection and human pose estimation respectively. The former often uses a pretrained object detector such as the Faster RCNN detector. The latter also uses a very deep backbone. So, one main drawback is the time complexity, which in turn motivates more efficient bottom-up approaches in the literature. For end-to-end top-down methods such as the Mask RCNN, the performance still lacks significantly behind (e.g., about 64 AP  vs 69.6 AP by our method in the COCO benchmark).  Those being said, we will add top-down prior arts which can achieve similar efficiency as our method in comparisons. Overall, we think the proposed method is of great significance to the field of human pose estimation given the best accuracy and near real-time efficiency obtained across the two benchmarks in experiments.
>
>
> ### 3. This paper lacks of ablation study to show the effect of proposed modules. It should clearly introduce where the improvement over the baseline come from. The following experiment results should be given: a) the effect of local-global contextual adaptation; b) the effect of Gaussian reweighting; c) the effect of convolution message passing module; d) The reviewer is also curious about the result of applying the local window search (the argmax) on the KAMs, which is similar to Table.1 Emp. Bound.
>
> **Ans**: Thank you for your suggestions. For a) and b), we perform a series of experiments for comprehensively studying the effect of each proposed modules. Please see Table 1 and Table 2 and the corresponding discussions. For c), the baseline center-offset method already utilizes a post-hoc absorbing scheme which  absorbs the regressed keypoint to the closest keypoint among the keypoints identified from the keypoint heatmaps (i.e., argmax). The empirical upper bound is computed by the local search w.r.t. the ground-truth, and the search-refined keypoints are mainly not local-max points, which motivates the proposed local-global contextual adaptation method to alleviate the gap between the baseline center-offset method and the empirical upper bound with state-of-the-art performance achieved.
>
> **Table 1. Ablation studies on the three components: the OKS loss, the Gaussian reweighing method for heatmaps and the Attentive Normalization.**
>
> | HRNet-W32 | OKS Loss | Reweigh | AttNorm |  AP  | AP$^{50}$ | AP$^{75}$ | AP$^M$ | AP$^L$ |
> | :-------: | :------: | :-----: | :-----: | :--: | :----: | :----: | :---: | :---: |
> | baseline  |    -     |    -    |    -    | 60.0 |  84.4  |  66.4  | 54.0  | 71.1  |
> |    (a)    |    ✔     |    -    |    -    | 66.1 |  86.7  |  72.7  | 60.0  | 75.6  |
> |    (b)    |    -     |    ✔    |    -    | 67.6 |  87.0  |  74.3  | 62.1  | 76.7  |
> |    (c)    |    ✔     |    ✔    |    -    | 69.0 |  87.0  |  75.2  | 63.4  | 77.5  |
> |    (d)    |    ✔     |    -    |    ✔    | 65.8 |  86.8  |  72.3  | 59.3  | 75.4  |
> |    (e)    |    -     |    ✔    |    ✔    | 67.5 |  86.6  |  74.1  | 62.2  | 76.7  |
> |    (f)    |    ✔     |    ✔    |    ✔    | 69.6 |  87.5  |  75.9  | 64.1  | 78.0  |
> |
>
> **Table 2. Ablation study of the different size of the local KEMs**
>
> | size of local KEMs | AP   | AP$^{50}$ | AP$^{75}$ | AP$^M$ | AP$^L$ | FPS  |
> | :----------------: | ---- | --------- | --------- | ------ | ------ | ---- |
> |    $7 \times 7$    | 68.4 | 86.6      | 74.9      | 63.4   | 76.6   | 21.8 |
> |   $11\times 11$    | 69.6 | 87.5      | 75.9      | 64.1   | 78.0   | 20.7 |
> |   $15\times 15$    | 69.3 | 87.1      | 75.2      | 63.2   | 78.3   | 16.5 |
> |   $19\times 19$    | 69.0 | 87.1      | 75.2      | 62.8   | 78.2   | 13.2 |
> |
>
> ### 4. The inference time of the proposed method increases with the number of persons, as mentioned above. The details of how to test the inference speed for multi-person images and the reasons why LOGO-CAP show superiority over the bottom up methods (DEKR[11]) on inference speed should be given.
>
> **Ans**: With all due respect, this seems to be a misunderstanding of the proposed method. Given an input image, regardless of how many person instances, the proposed method will compute the final human pose estimation results in a single forward computation in inference, exactly like the DEKR method without needing the post-hoc scoring procedure. This inference procedure is clearly presented in Section 3.2.1 (Eqn.3 to Eqn.10).  We show the breakdown of inference time of the proposed method in Table 3.
>
> **Table 3: The breakdown of inference time of the proposed LOGO-CAP method.  The inference time is averagelycomputed using a NVIDIA TITAN RTX GPU on the COCO-val-2017 dataset. For each model, we seperately report the averaged inference time across 5000 images, the averaged inference time in the images that detect only one person, the avaraged inference time in the images that have 30 persons.**
>
> |                | Number of Persons | Backbone | Local KEMs | Local KAMs | Global KAMs |
> | :------------: | :---------------: | :------: | :--------: | ---------- | ----------- |
> | LOGO-CAP (W32) |         -         | 38.6 ms  |  3.05 ms   | 2.49 ms    | 2.85 ms     |
> |                |         1         |          |  2.39 ms   | 1.14 ms    | 1.12 ms     |
> |                |        30         |          |  3.69 ms   | 3.49 ms    | 5.87 ms     |
> |
> | LOGO-CAP (W48) |         -         | 99.9 ms  |  4.18 ms   | 3.00 ms    | 3.34 ms     |
> |                |         1         |          |  3.19 ms   | 1.10 ms    | 1.07 ms     |
> |                |        30         |          |  2.97 ms   | 3.59 ms    | 5.97 ms     |
> |
>
> For measuring the inference speed, we also clearly present the method in Section 4.3, e.g., L.342-344 ``In comparing the inference speed, we test all the models on a single TITAN RTX GPU for its popularity in practice. The average inference speed, FPS (frames per second), over the 5000 images in COCO-val-2017 is used for the comparison."
>
> Furthermore, for why the proposed LOGO-CAP shows superiority over the DEKR on inference speed, we also explicitly present that ``Comparing with DEKR, the speed improvement of our LOGO-CAP is from the lightweight design of head modules since the same backbones are used." in L. 349-350.  It is also noted that the original source code of the DEKR runs slower than our implementation by large margins, and we try our best to ensure the fairness of the comparison at the algorithmic level (L. 344-345).

---

> > ### Author Response · Authors · 2021-08-10
> > **Reply: Part 2**
> >
> > ### 5. The inaccuracy of global KEMs comes from the global keypoint heatmaps, while the KAMs measure the error of center-offset results. Here is the mismatch between heatmaps and center-offsets. How can KAMs help refine global KEMs?
> >
> > **Ans**: Indeed, both global KEMs and local KAMs have uncertainties, and they can not be easily handled individually. So, in the proposed method, local KAMs and global KEMs are connected via the local-global contextual adaptation in a fully differentiable way. Then, the OKS loss term enable positively and jointly refining both of them in the supervised training, together with the heatmap and offset field loss terms (see Section 3.2.2 in the submission). The fully differentiable local-global contextual adaption module and the OKS guided supervised training are the key. To quantitatively compare these, we compare the performance for using KEMs only, using KAMs only and the proposed LOGO-CAP method in Table 4 respectively. It shows the effectiveness of the proposed method.
> >
> > **Table 4. Ablation study of using different type of the priors for our LOGO-CAP. The KEMs and KEMs denote the global ones used for the adaptation.**
> >
> > | Type of the Prior      | AP   | AP$^{50}$ | AP$^{75}$ | AP$^M$ | AP$^L$ |
> > | ---------------------- | ---- | --------- | --------- | ------ | ------ |
> > | KEMs only              | 59.4 | 80.8      | 62.8      | 50.9   | 71.6   |
> > | KAMs only              | 65.7 | 86.0      | 72.3      | 60.6   | 74.0   |
> > | LOGO-CAP (KEMs + KAMs) | 69.6 | 87.5      | 75.9      | 64.1   | 78.0   |
> >
> > ### 6. The writing can be improved. Notations in this paper are not well defined. There are also too many similar concepts with different names, e.g. Local heatmaps, local KEMs, KAMs. It would be better if they use a consistent name throughout the paper. Some important details of the proposed approach are missing (e.g., how to compute the geometric mesh of the global KEMs
> >
> > **Ans**: Thank you for your suggestion, we will revise our paper with better organized notations. For computing the geometric mesh, we provide the pseudo code, Algorithm 1 in the supplementary. Basically, the computing of the geometric mesh is  simple with two line PyTorch/Python codes: (1) using the Torch meshgrid function to get the standard mesh grid from $-48$ to $48$ with the step size of 1, and (2) adding the standard mesh grid with the tensor broadcasting to the initial center-offset estimations. We will release our code.

---

> > > ### Comment · Reviewer_k2Ew · 2021-08-23
> > >
> > > Thank you for your reply. I have carefully read the rebuttals and the comments from other reviewers. The authors have addressed some of my concerns in the reply. However, I still have some major concerns.
> > >
> > > 1.	Top-down vs bottom-up, computational complexity
> > >
> > > a)	From my point of view, the proposed method is not a typical “bottom-up” approach. One major drawback of the top-down approaches is that their computational cost increases with the number of people (This is also pointed out by the authors in Introduction). But the proposed method (LOGO-CAP) also suffers from such problem. The time complexity of applying the convolution operator is $O(P \cdot K \cdot \omega^2 \cdot H^2)$. So I think it is more like a top-down approach, and it should be compared with other top-down approaches.
> > >
> > > b)	I would like to suggest adding such a baseline method [BASELINE]: Simply using the predicted offset fields to get an initial pose, and then obtain a bounding box for each person given the initial pose. Then, for each person, run a single person human pose estimation network (e.g. HRNet). This [BASELINE] might also produce good enough results.
> > >
> > > c)	Some may argue that in LOGO-CAP the computation cost for each person is rather small compared to the backbone network. And comparing it with a single person human pose estimation network (e.g. HRNet) is not fair enough. But I would like to point out that the single person human pose estimation network can be very small too. Especially in many real applications, the model can even run at less than 1ms. Personally speaking, the actual runtime is about implementation & engineering, and the theoretical complexity is what we care about.
> > >
> > > 2.	About experiments
> > >
> > > First of all, I would like to thank the authors for adding some ablation experiments. In the authors’ responses, it reads that “The ablation studies verify the effectiveness of integrating the three components (OKS loss, Gaussian reweighting, and Attentive Normalization) in the proposed method.” In Table1, we see that OKS loss and Gaussian reweighting are very important for achieving good results. These techniques have been studied in literature, so they are not the contributions of this paper.
> > > There are still some concerns. The major problem of the experiments is that the effectiveness of the proposed components is not well-validated to support the claims in the paper.
> > >
> > > a)	Line 81-83, it introduces a simple baseline, what is the result of that?
> > >
> > > b)	Line 87, it reads “the structural relationship between different keypoints of a human pose needs to be taken into account”. If I understand correctly, the structural relationship is encoded by the convolutional message passing (CMP). But the effectiveness of CMP is not validated. One way to do so is to replace CMP with a simple 1x1 convolution, which is used to decrease the feature dimension.
> > >
> > > c)	In Table 4 (authors’ reply), global KEMs do not perform as good as KAMs (59.4 vs 65.7). (1) In the paper, especially Figure 1, it seems that KAMs are used to refine KEMs. If KEMs are far worse than KAMs, why not improve upon KAMs?
> > >
> > > d)	I am also curious about the results of global KEMs + Gaussian reweighting. Does it already achieve good results?

---

> > > > ### Author Response · Authors · 2021-08-24
> > > > **Response to the new comments: Part1**
> > > >
> > > > **We thank you for your new comments. We hope that our new responses can address your remained concerns.**
> > > >
> > > > **Q1.a)** From my point of view, the proposed method is not a typical “bottom-up” approach. One major drawback of the top-down approaches is that their computational cost increases with the number of people (This is also pointed out by the authors in Introduction). But the proposed method (LOGO-CAP) also suffers from such a problem. The time complexity of applying the convolution operator is ​. So I think it is more like a top-down approach, and it should be compared with other top-down approaches.
> > > >
> > > > **A1.a):** We would like to argue on this point again with elaborated discussions.
> > > >
> > > > State-of-the-art top-down approaches often use cropped single-person image patches as inputs in both training and inference, where single-person image patches are computed based on a pretrained object detector such as the Faster-RCNN. They are clearly top-down due to this information flow.
> > > >
> > > > The proposed method and other grouping-based bottom-up approaches (OpenPose, PifPaf and Associative embedding, DEKR etc.) take an entire image as input in both training and inference, and directly regress the keypoints with different grouping/refining processes. Such an information flow distinguishes them from the aforementioned top-down approaches. The grouping/refining processes are typically person centered, which makes the computational cost increase with the number of people, but in a light-weight way.
> > > >
> > > > More specifically, we would like to point out four aspects as follows.
> > > >
> > > > First of all, the time complexity ​ can be rewritten as ​$\mathcal{O}(cPK)$ because the parameters ​ $\omega$ and ​$H$ are fixed as constants. Considering that a given input image has ​ persons and we need to estimate ​ body parts for each person of ​ persons, it is necessary to use ​ times to get the final results no matter how clever the used method or the observer is, unless you can precisely guess the number of persons without seeing the image.
> > > >
> > > > Secondly, if our work is treated as more like a top-down approach, the sentence of “the grouping-based bottom-up approaches are also more like top-down approaches” should also be true because the great deal of works, OpenPose, PifPaf and Associative Embedding, were designed to group people one-by-one from the part-level information. If we assume that the grouping operations by those approaches only require a small constant times ​ for each limb of each person and the number of limbs are linear related to the number of keypoints, their time complexity should also be ​.
> > > >
> > > > Thirdly, if our work is treated as a top-down approach, its headnet should be a point-wised and lightweight network, which is totally different from the existing top-down approaches that estimate human poses from the full contexts of image. A possible comparison is provided for your concern 1. c).
> > > >
> > > > Lastly, we would like to quote the comment “From my personal point of view, current top-down pose estimation methods are relatively mature and the main way to enhance performance is to improve the backbone. While bottom-up approaches rely more on design because of the problem of ‘how to group people’” by Review 1GBc. As we implement our headnet in the minimal design principle, our LOGO-CAP would not be a top-down approach that claims contributions on improving the backbone.
> > > >
> > > > In summary, we keep our assertion that our proposed method is a good, fast, fully end-to-end, and bottom-up approach. With all due respect, we thank you for your concern, which motivates us to comprehensively study the true top-down approach from the perspective of local-global contextual adaptation.
> > > >
> > > > **Q1.b)** I would like to suggest adding such a baseline method [BASELINE]: Simply using the predicted offset fields to get an initial pose, and then obtain a bounding box for each person given the initial pose. Then, for each person, run a single person human pose estimation network (e.g. HRNet). This [BASELINE] might also produce good enough results.
> > > >
> > > > **A1.b)**: Yes, such a [BASELINE] does produce good results. To better respond to this point, we run the official top-down approach HRNet-W32 on the COCO-val-2017 dataset with 256x192 input size for different sources of the input bounding boxes:
> > > >
> > > > (i). the boxes are generated from the initial poses (with W32-backbone)
> > > >
> > > > (ii) the boxes are generated from the pose parameters of LOGO-CAP-W32 model
> > > >
> > > > (iii) the boxes are generated from the pose parameters of DEKR-W32 model
> > > >
> > > > (iii) the officially-used person detection results by HRNet
> > > >
> > > > (iv) the ground-truth bounding boxes of COCO-val-2017 dataset.
> > > >
> > > > We report the performance of running the suggested top-down [BASELINE] against the number of input bounding boxes from different sources in the attached table.
> > > >
> > > >
> > > > | Source of Bboxes          | \# Boxes | AP   | AP$^{50}$ | AP$^{75}$ | AP$^M$ | AP$^L$ |
> > > > | ------------------------- | -------- | ---- | --------- | --------- | ------ | ------ |
> > > > | (i) initial center-offset | 46820    | 72.4 | 87.7      | 79.2      | 68.7   | 78.4   |
> > > > | (ii) LOGO-CAP-W32         | 40423    | 72.5 | 87.7      | 79.0      | 68.2   | 79.1   |
> > > > | (iii) DEKR-W32            | 36192    | 72.3 | 88.0      | 78.8      | 68.1   | 78.8   |
> > > > | (iv) Detection            | 104125   | 74.4 | 90.5      | 81.9      | 70.8   | 81.0   |
> > > > | (v) GT boxes              | 6352     | 76.5 | 93.5      | 83.7      | 73.9   | 80.8   |
> > > >
> > > > As shown in the table, applying one more stage with a larger backbone for the bottom-up approaches (including the baseline, our LOGO-CAP and DEKR) could improve the performance, however, they are still inferior to the top-down approach (i.e, iv in Table 1 below) by large margins. A major reason is that the object detectors will provide 100k bounding boxes, which is 2.5x larger than the bottom-up sources. In this perspective, a key reason for why the bottom-up approaches are worth studying is that they can accurately and directly estimate human pose parameters without incurring information from the extra and redundant bbox proposals.
> > > >
> > > > For the compared bottom-up approaches, their performance under the suggested [BASELINE] is very similar. Although the final performance can be improved by this [BASELINE], they are still far lower than the empirical upper bound presented in Table 1 of our original submission.
> > > >
> > > > Furthermore, when applying the GT boxes, the top-down approach still remains a large gap from our reported empirical upper bound in Table 1 in the submission. It is thus promising to study more on the center-offset end-to-end learning.
> > > >
> > > > **Q1.c)** Some may argue that in LOGO-CAP the computation cost for each person is rather small compared to the backbone network. And comparing it with a single person human pose estimation network (e.g. HRNet) is not fair enough. But I would like to point out that the single person human pose estimation network can be very small too. Especially in many real applications, the model can even run at less than 1ms. Personally speaking, the actual runtime is about implementation & engineering, and the theoretical complexity is what we care about.
> > > >
> > > > **A1.c)**: We can directly compare our LOGO-CAP with the small single-person pose estimation methods per your suggestion “the single person human pose estimation network can be very small too”. We use the most up-to-date SOTA approach, Lite-HRNet (published in CVPR 2021) to make a comparison on the COCO-val-2017 dataset.  For each model, we report the 1-person inference time on a single NVIDIA TITAN RTX GPU. The batch-size is set to 1 to exclude the influence of the batchmode. For our LOGO-CAP, we report the time for the headnet/backbone respectively. For your viewpoint of “the actual runtime is about implementation & engineering”, we would like to emphasize that engineering a natively-fast method to be faster is usually easier than engineering a natively-slower one. For the comparison of runtime, we actually tried our best to keep the fairness on the same hardware.
> > > >
> > > > | model                   | AP   | AP$^{50}$ | AP$^{75}$ | AP$^M$ | AP$^L$ | 1-person inference time |
> > > > | ----------------------- | ---- | --------- | --------- | ------ | ------ | ----------------------- |
> > > > | Lite-HRNet-18 (256x192) | 64.8 | 86.7      | 73.0      | 62.1   | 70.5   | 46.51ms                 |
> > > > | Lite-HRNet-18 (384x288) | 67.6 | 87.8      | 75.0      | 64.5   | 73.7   | 48.07ms                 |
> > > > | Lite-HRNet-30 (256x192) | 67.2 | 88.0      | 75.0      | 64.3   | 73.1   | 79.36ms                 |
> > > > | Lite-HRNet-30 (384x288) | 70.4 | 88.7      | 77.7      | 67.5   | 76.3   | 82.64ms                 |
> > > > | LOGO-CAP-W32            | 69.6 | 87.5      | 75.9      | 64.1   | 78.0   | 4.65ms/38.6ms           |
> > > > | LOGO-CAP-W48            | 72.2 | 88.9      | 78.9      | 68.1   | 78.9   | 5.36ms/99.9ms           |

---

> > > > > ### Author Response · Authors · 2021-08-24
> > > > > **Response to the new comments: Part2**
> > > > >
> > > > >
> > > > > **Q2.a)** Line 81-83, it introduces a simple baseline, what is the result of that?
> > > > >
> > > > > **A2.a):** We think we have reported it in Table 1 (a) of the rebuttal, which is corresponding to the description “Motivated by the above observation, a straightforward way is just to learn a local heatmap (11 ×11) for each human pose keypoint based on the learned center and offset vectors, and then to compute the refined keypoints by taking arg max within the local heatmap.”
> > > > >
> > > > > **Q2.b)** Line 87, it reads “the structural relationship between different keypoints of a human pose needs to be taken into account”. If I understand correctly, the structural relationship is encoded by the convolutional message passing (CMP). But the effectiveness of CMP is not validated. One way to do so is to replace CMP with a simple 1x1 convolution, which is used to decrease the feature dimension.
> > > > >
> > > > > **A2.b)**: We need 48 hours to train the network per your suggestion. We will reply to you separately once the network is trained.
> > > > >
> > > > > **Q2.c)** In Table 4 (authors’ reply), global KEMs do not perform as good as KAMs (59.4 vs 65.7). (1) In the paper, especially Figure 1, it seems that KAMs are used to refine KEMs. If KEMs are far worse than KAMs, why not improve upon KAMs?
> > > > >
> > > > > **A2.c):** Since the global KEMs are actually the standard Gaussian around each initial keypoint, it cannot provide more information for refinement. When we enforce the use of local KAMs for adaptation with only global KEMs, the uncertainty from local KEMs will affect the results. That is the reason why only using global KEMs is worse than using global KAMs.
> > > > >
> > > > > **Q2.d) **I am also curious about the results of global KEMs + Gaussian reweighting. Does it already achieve good results?
> > > > >
> > > > > **A2.d):** We think we have run the experiments in Table 4 of the rebuttal. It only achieved an AP of 65.7, that is not good enough.

---

> > > > > > ### Author Response · Authors · 2021-08-26
> > > > > > **Ablation Study for CMP**
> > > > > >
> > > > > > Thanks for your suggestion again. We have trained our LOGO-CAP-W32 model by replacing the convolutional message passing with a simple 1x1 convolution for decreasing the feature dimension. The configuration of hyperparameters is the same as the previously used models. The results are reported in the following table, which shows that the convolutional message passing module obtains an absolute improvement of AP by 0.8%.
> > > > > >
> > > > > > |         |  AP [%]  | AP$^{50}$ [%] | AP$^{75}$ [%] | AP$^M$ [%] | AP$^L$ [%] |
> > > > > > |:-------:|:----:|:---------:|:---------:|:------:|:------:|
> > > > > > | w/ CMP  | 68.8 |   86.9    |   74.9    |  63.1  |  77.5  |
> > > > > > | w/o CMP | 69.6 |   87.5    |   75.9    |  64.1  |  78.0  |
> > > > > >
> > > > > > We will update this ablation study to our paper.

---

> > > > > ### Comment · Reviewer_k2Ew · 2021-09-10
> > > > > **The submission needs major revision**
> > > > >
> > > > > Thanks for your reply. During the rebuttal, the authors did a lot of work. They added many new experimental results, which are critical but missing in the original submission. However, I am still a little bit concerned that the proposed method is not a typical “bottom-up” approach. And I believe it is reasonable to compare with both top-down and bottom-up approaches.
> > > > >
> > > > > In total, I agree with Reviewer MSva that the original submission still needs major revision before being published. All discussions and results during the rebuttal should be added into the paper.

---

> > > > > > ### Author Response · Authors · 2021-09-10
> > > > > > **Thanks for your reply.**
> > > > > >
> > > > > > We thank you for your reply.
> > > > > >
> > > > > > We would like to point out that all the experiments in the rebuttal strongly support the results in the submission. For the suggestions of comparing with top-down approaches, we may add a full comparison in the supplemental material due to the page limit.

---

### Official Review · Reviewer_MSva · 2021-07-16

**Rating:** 4
**Confidence:** 5

**Summary:**

This paper proposed a local-global contextual adaptation method for the bottom-up human pose estimation.  In the proposed method, local keypoint expansion maps (KEMs) and global KEMs are both extracted. The pose-sensitive kernels (keypoint attraction maps, KAMs) are learned from the local KEMs and then performed to the global KEMs. The final prediction is generated from the refined global KEMs. The proposed method is end-to-end trainable and achieves state-of-the-art performance on COCO and OCHuman datasets.


**Ethical Concerns:**

No ethical issues found.

**Limitations And Societal Impact:**

Yes.

**Main Review:**

Pros:
- This paper proposed an interesting way to refine the results of center-offset-based pose estimation.
- The proposed method outperforms the state of the art on COCO and OCHuman datasets.
- The details of the proposed method are well described.

Cons:
-The experiment is not sufficient to prove the effectiveness of the proposed method. More ablation studies and analyses should be provided.
1)  The KAM is the key component of the proposed method, so it is better to investigate different ways for KAM generation. For example, different sizes of local KEMs, different designs of convolutional message passing module, use or do not use KAM to refine the global KAMs.
2) Some proposed modules should be evaluated in the ablation studies, e.g. the CMP, the attention normalization and the effectiveness of performing KAM to global KEMs, etc.
3)  It is important to show the inference speed comparison between the baseline and the proposed LOGO-CAP.
4) Further analysis should be provided. What kinds of KAMs kernels are generated adaptively and how do the KAMs refine the global maps? What kind of hard cases are refined by the proposed method.

-This paper claims that ‘this is the first fully end-to-end trainable method that achieves the state-of-the-art performance’ (in L. 161). However, many end-to-end trainable pose estimation methods have been proposed, e.g. Mask R-CNN, ICCV 2017,  HGG[13].

-The paper should be further polished before being published.
1) More explanations should be presented in the caption of each figure, e.g. Figure 1, 3, 4.
2) In Figure 4, the major difference between the initial offsets and the refined pose should be highlighted. It is hard to find the improvements.
3) Typos:
In L.45, ‘approach is ’ -> ‘approaches are’
In L. 280, the citation is missed.
In L. 319, 321, 328, ‘points’ should be ‘%’.


**Time Spent Reviewing:**

5 hours

---

> ### Author Response · Authors · 2021-08-10
> **Reply**
>
> Thanks for your review. We will further polish our paper. For your major concerns, we provide detailed responses as follow.
>
> ### 1. The KAM is the key component of the proposed method, so it is better to investigate different ways for KAM generation. For example, different sizes of local KEMs, different designs of convolutional message passing module, use or do not use KAM to refine the global KAMs. Some proposed modules should be evaluated in the ablation studies, e.g. the CMP, the attention normalization and the effectiveness of performing KAM to global KEMs, etc.
>
> **Ans**: Thank you for your suggestions. First, without using the KAMs to refine the global KEMs, it is the center-offset baseline with the performance reported in the Table 1 in the submission (60.1 AP by the baseline and 70.0 AP by the proposed method). Second, for different sizes of local KEMs, we select $11\times 11$ as the kernel size in experiments based on the empirical observations in computing the Table 1. Following your suggestions, we perform an ablation study with the results shown in Table 2, which confirm that the $11\times 11$ kernel size obtains the best performance. One possible explanation is that smaller kernel sizes fail to compensate the uncertainty of the initial keypoint estimation results, while larger kernel sizes may introduce more nuisances factors that affect the performance, such as the ``collision" between different local KEMs of different keypoints from either the same person or adjacent different persons. Third, for different designs of the CMP module, we perform the ablations study between with and without using the attentive normalization module in Table 1. The attentive normalization module is introduced to account for the underlying structural and appearance variations of different keypoints in the convolutional messaging passing computation in a light-weight way. As mentioned in the submission, more sophisticated and computationally expensive methods such as the Transformer models could also be exploited. We leave this to future work since one main goal of this paper is to develop efficient human pose estimation methods by exploiting minimally-simple designs.
>
> **Table 1. Ablation studies on the three components: the OKS loss, the Gaussian reweighing method for heatmaps and the Attentive Normalization.**
>
> | HRNet-W32 | OKS Loss | Reweigh | AttNorm |  AP  | AP$^{50}$ | AP$^{75}$ | AP$^M$ | AP$^L$ |
> | :-------: | :------: | :-----: | :-----: | :--: | :----: | :----: | :---: | :---: |
> | baseline  |    -     |    -    |    -    | 60.0 |  84.4  |  66.4  | 54.0  | 71.1  |
> |    (a)    |    ✔     |    -    |    -    | 66.1 |  86.7  |  72.7  | 60.0  | 75.6  |
> |    (b)    |    -     |    ✔    |    -    | 67.6 |  87.0  |  74.3  | 62.1  | 76.7  |
> |    (c)    |    ✔     |    ✔    |    -    | 69.0 |  87.0  |  75.2  | 63.4  | 77.5  |
> |    (d)    |    ✔     |    -    |    ✔    | 65.8 |  86.8  |  72.3  | 59.3  | 75.4  |
> |    (e)    |    -     |    ✔    |    ✔    | 67.5 |  86.6  |  74.1  | 62.2  | 76.7  |
> |    (f)    |    ✔     |    ✔    |    ✔    | 69.6 |  87.5  |  75.9  | 64.1  | 78.0  |
> |
>
> **Table 2. Ablation study of the different size of the local KEMs**
>
> | size of local KEMs | AP   | AP$^{50}$ | AP$^{75}$ | AP$^M$ | AP$^L$ | FPS  |
> | :----------------: | ---- | --------- | --------- | ------ | ------ | ---- |
> |    $7 \times 7$    | 68.4 | 86.6      | 74.9      | 63.4   | 76.6   | 21.8 |
> |   $11\times 11$    | 69.6 | 87.5      | 75.9      | 64.1   | 78.0   | 20.7 |
> |   $15\times 15$    | 69.3 | 87.1      | 75.2      | 63.2   | 78.3   | 16.5 |
> |   $19\times 19$    | 69.0 | 87.1      | 75.2      | 62.8   | 78.2   | 13.2 |
> |
>
> ### 2.  It is important to show the inference speed comparison between the baseline and the proposed LOGO-CAP.
>
> **Ans**: Thank you for your suggestion. Compared with the baseline, the inference time overhead by the proposed method is small. With the HRNet-W32 backbone, the baseline obtains 23.1 FPS (with 60.0 AP on the full COCO-val-2017 dataset) while our LOGO-CAP achieves 20.7 FPS (with 69.6 AP). More specifically, the proposed LOGO-CAP method only consumes about 10ms in inference after the computation of the feature backbone. We show the breakdown of inference times in Table 3.
>
> **Table 3: The breakdown of inference time of the proposed LOGO-CAP method.  The inference time is averagelycomputed using a NVIDIA TITAN RTX GPU on the COCO-val-2017 dataset. For each model, we seperately report the averaged inference time across 5000 images, the averaged inference time in the images that detect only one person, the avaraged inference time in the images that have 30 persons.**
>
> |                | Number of Persons | Backbone | Local KEMs | Local KAMs | Global KAMs |
> | :------------: | :---------------: | :------: | :--------: | ---------- | ----------- |
> | LOGO-CAP (W32) |         -         | 38.6 ms  |  3.05 ms   | 2.49 ms    | 2.85 ms     |
> |                |         1         |          |  2.39 ms   | 1.14 ms    | 1.12 ms     |
> |                |        30         |          |  3.69 ms   | 3.49 ms    | 5.87 ms     |
> |
> | LOGO-CAP (W48) |         -         | 99.9 ms  |  4.18 ms   | 3.00 ms    | 3.34 ms     |
> |                |         1         |          |  3.19 ms   | 1.10 ms    | 1.07 ms     |
> |                |        30         |          |  2.97 ms   | 3.59 ms    | 5.97 ms     |
> |
>
>
>
> ### 3. Further analysis should be provided. What kinds of KAMs kernels are generated adaptively and how do the KAMs refine the global maps? What kind of hard cases are refined by the proposed method.
>
> **Ans**:  The KAMs kerenels are computed based on Eqn.7 in the submission with the detailed pseudo-code provided in Algorithm 1 in the supplementary material. Please refer to the supplementary for more details, and we will try to include it in the main text in revision. As defined in Eqn.9 in the submission, the learned KAMs is used to convolve the cropped global heatmaps to facilitate local-global contextual adaptation. The resulting performance improvement justifies its effectiveness. More specifically, for hard cases that are refined by the proposed method, we show examples in both Fig.1 and Fig. 4 in the submission. In the left of Fig. 1, the initial center-offset regression result for the squatting woman is inaccurate and only obtains an OKS of 66.1\%, which is then refined by our proposed local-global contextual adaptation to 80.4\%. In Fig.4, we also report the OKS difference between the initial and the final estimations. Quantitatively, Table 1 in the submission clearly shows the significance  (10\% absolution AP improvement) of the proposed method compared with the baseline.
>
>
> ### 4. This paper claims that ‘this is the first fully end-to-end trainable method that achieves the state-of-the-art performance’ (in L. 161). However, many end-to-end trainable pose estimation methods have been proposed, e.g. Mask R-CNN, ICCV 2017, HGG[13].
>
> **Ans**: With all due respect, we clearly state the contribution under the context that "The proposed LOGO-CAP makes three main contributions to the field of bottom-up human pose estimation" (L. 158-159). We are aware of the great work of Mask R-CNN and HGG. We compared with HGG in the Table 2 of the submission. Both of them obtained significantly inferior performance than the proposed method. That being said, we understand your concern and will revise the claim to be more precise: It achieves state-of-the-art bottom-up human pose estimate performance. It is fully end-to-end trainable, unlike many other bottom-up human pose estimation methods which entail separate post-hoc processing to improve performance.

---

> > ### Comment · Reviewer_MSva · 2021-08-23
> > **Feedback to the rebuttal**
> >
> > Thanks for your reply.
> >
> > I have read the feedback and some of my concerns have been addressed, e.g. the ablation study, the inference speed, etc. However, some questions still exist.
> > To prove the effectiveness of local KAMs, it seems better to compare the proposed method with the 'KEM only' method instead of the baseline in Table 1 of the original submission.  The 'KEM only' model may have a similar parameter number and does not include the KAM module.
> >
> > It is confused that the 'KEM only' model does not outperform the baseline, which has fewer parameters and is without heatmap supervision.
> >
> > The proposed method 'AttNorm' decreases the results in two comparisons, i.e.  (a) vs (d) and (b) vs (e), and only improves the results in the comparison of (c) vs (f). What is the root cause of these results?
> >
> > Above all, I think the authors provide many helpful results in the rebuttal, but the original submission still needs major revision before being published. All the results need to be added to the submission and the writing needs to be further revised. I tend to keep the original rating now.

---

> > > ### Author Response · Authors · 2021-08-24
> > > **Response to your feedback**
> > >
> > > **We sincerely thank you for your constructive comments. Here is our new responses.**
> > >
> > > **Q1:** To prove the effectiveness of local KAMs, it seems better to compare the proposed method with the 'KEM only' method instead of the baseline in Table 1 of the original submission. The 'KEM only' model may have a similar parameter number and does not include the KAM module.
> > >
> > >
> > >
> > > **A1:** Thanks for your suggestion. We have already added it in Table 1 of the original submission.
> > >
> > >
> > >
> > > **Q2:** It is confusing that the 'KEM only' model does not outperform the baseline, which has fewer parameters and is without heatmap supervision.
> > >
> > > **A2:** Since the global KEMs are actually the standard Gaussian around each initial keypoint, it cannot provide more information for refinement. When we enforce the use of local KAMs for adaptation with only global KEMs, the uncertainty from local KEMs will affect the results. That is the reason why only using global KEMs is worse than using global KAMs. We are also interested in learning accurate human poses without using heatmaps in our future work.
> > >
> > >
> > >
> > > **Q3:** The proposed method 'AttNorm' decreases the results in two comparisons, i.e. (a) vs (d) and (b) vs (e), and only improves the results in the comparison of (c) vs (f). What is the root cause of these results?
> > >
> > > **A3:** As the AttNorm exploits the feature recalibration mechanism for Neural Networks, we resorted to use this characteristic to address the inaccurate local KEMs estimation when we found the fascinating empirical performance upper bounding for center-offset regression results. However, such a design did not bring any improvements. When we went further on human pose estimation, we found that although the global heatmaps are not that accurate, they still provide meaningful information for this task. Then, we used AttNorm and found that the feature recalibration mechanism of AttNorm works with two different information sources as shown in the ablation study (Table 1 of the last rebuttal). For those two sets of ablation studies, they demonstrated that the feature recalibration mechanism of AttNorm requires two complementary information sources instead of any single one.
> > >
> > >
> > >
> > > **Q4:** Above all, I think the authors provide many helpful results in the rebuttal, but the original submission still needs major revision before being published. All the results need to be added to the submission and the writing needs to be further revised. I tend to keep the original rating now
> > >
> > > **A4:** We have revised our manuscript after replying to all the concerns by different reviewers. We sincerely thank you for the constructive comments.

---

### Decision · Program_Chairs · 2021-09-27

**Decision:**

Reject

**Comment:**

This submission received 3 strongly diverging final ratings: 4, 3, 8.
On the positive side, reviewers appreciated the central idea and strong empirical performance.
At the same time, most reviewers felt like ablations were insufficient, some of the claims were inaccurate, and overall the submission needed another major revision and a round of peer reviews before it can be considered for publication.
After an extensive discussion between the authors and the reviewers during the rebuttal period, the ratings remained unchanged.
Overall, the AC agrees with the skeptical reviewers that the weaknesses of this submission at this point outweigh its strengths, so the final recommendation is to reject.